# Single amino acid change alters specificity of the multi-allelic wheat stem rust resistance locus *SR9*

Jianping Zhang [1,2,3,15], Jayaveeramuthu Nirmala[4,15], Shisheng Chen [5,15], Matthias Jost[1,15], Burkhard Steuernagel [6], Mirka Karafiatova [7], Tim Hewitt [1], Hongna Li [5], Erena Edae[4], Keshav Sharma[8], Sami Hoxha[2], Dhara Bhatt[1], Rea Antoniou-Kourounioti [9], Peter Dodds [1], Brande B. H. Wulff [10,11], Jaroslav Dolezel [7], Michael Ayliffe[1], Colin Hiebert[12], Robert McIntosh[2], Jorge Dubcovsky [13,14], Peng Zhang [2] ✉, Matthew N. Rouse[4,8] ✉ & Evans Lagudah [1,2] ✉

Most rust resistance genes thus far isolated from wheat have a very limited number of functional alleles. Here, we report the isolation of most of the alleles at wheat stem rust resistance gene locus *SR9*. The seven previously reported resistance alleles (*Sr9a*, *Sr9b*, *Sr9d*, *Sr9e*, *Sr9f*, *Sr9g*, and *Sr9h*) are characterised using a synergistic strategy. Loss-of-function mutants and/or transgenic complementation are used to confirm *Sr9b*, two haplotypes of *Sr9e* (*Sr9e_h1* and *Sr9e_h2*), *Sr9g*, and *Sr9h*. Each allele encodes a highly related nucleotide-binding site leucine-rich repeat (NB-LRR) type immune receptor, containing an unusual long LRR domain, that confers resistance to a unique spectrum of isolates of the wheat stem rust pathogen. The only SR9 protein effective against stem rust pathogen race TTKSK (Ug99), SR9H, differs from SR9B by a single amino acid. SR9B and SR9G resistance proteins are also distinguished by only a single amino acid. The *SR9* allelic series found in the B subgenome are orthologs of wheat stem rust resistance gene *Sr21* located in the A subgenome with around 85% identity in protein sequences. Together, our results show that functional diversification of allelic variants at the *SR9* locus involves single and multiple amino acid changes that recognize isolates of wheat stem rust.

International efforts toward genetic resistance breeding for controlling stem rust disease caused by *Puccinia graminis* f. sp. *tritici* (*Pgt*) in wheat (*Triticum* spp.) started in the early 20th century[1–4]. Since then, many resistance (R) genes effective to wheat stem rust (*Sr* genes) have been genetically defined and deployed to combat the stem rust disease[5]. Sixthteen wheat *Sr* genes have been cloned to date by either map-based cloning (*Sr13*, *Sr21*, *Sr33*, *Sr35*, *Sr50*, *Sr55*, *Sr57*, *Sr60*, and *Sr62*) or target sequence capture approaches, including MutRenSeq (*Sr22*, *Sr26*, *Sr27*, *Sr45*, and *Sr61*), MutChromSeq (*Sr43*) and AgRenSeq (*Sr46*)[6–19]. *Sr55* and *Sr57* are pleiotropic resistance genes also conferring

adult plant resistance to wheat leaf rust, stripe rust, and powdery mildew. *Sr55* encodes an altered hexose transporter whereas *Sr57* encodes an ABC transporter protein[11,12]. *Sr60* and *Sr62* encode proteins with two putative kinase domains, and *Sr43* encodes a protein kinase fused to two domains of unknown function. The other 11*Sr* genes were demonstrated to encode nucleotide-binding leucine-rich repeat (NLR) immune receptor proteins and most of them have a predicted coiled-coil (CC) domain (CNL) at their N termini. Cryo-electron microscopy (EM) structure studies of plant NLRs changes upon detection of pathogen effectors that trigger immune responses have provided

insights into activated immune receptor function[20–25]. In the eudicot model species, *Arabidopsis thaliana*, indirect recognition of a pathogen effector by the CNL ZAR1 immune receptor induces the formation of a protein complex assembled as a pentamer, referred to as the ZAR1 resistosome, which functions as a calcium-permeable cation channel implicated in ZAR1 mediated immunity[20,26]. Similarly, Cryo-EM structure analysis showed that the wheat Sr35 NLR binds directly to the pathogen effector AvrSr35 and forms a pentameric Sr35-AvrSr35 resistosome complex that displays a non-selective calcium channel[9,24,25,27].

Characterization of the *SR9* locus began as early as 1956 by Knott and Anderson with the most recently described allele, *Sr9h*, designated in 2014[28–30]. The study of the inheritance of stem rust resistance began in the 1950s following a series of papers on flax rust by Flor[31]. These studies enabled R genes to be distinguished by differential responses to arrays of pathogen isolates, and be assigned to different chromosomes using aneuploid stocks[32]. The development of near-isogenic lines (NILs) for different resistance genes in a common genetic background allowed closer comparison of individual genes/alleles[29,33] and enabled the comparison of host responses in different laboratories worldwide since pathogen isolates were not commonly exchanged due to biosecurity risks. One of these studies[29] enabled the distinction of host responses of the original gene named *Sr9* from a gene named *SrKb1* that occurred at the same genetic locus, leading to their designations as the *Sr9a* and *Sr9b* alleles respectively[34]. The name *Sr9c* was retained for a resistance gene transferred to wheat from *Triticum timopheevii* but was later dropped[35]. A gene originally transferred from cultivated emmer (*T. dicoccum* cv. Yaroslav) by McFadden, and later named *Sr1* was postulated as an additional *Sr9* allele that was renamed *Sr9d*[28].

A gene named *Srv*[36] and *Srd1v*[37] from tetraploid wheat and transferred later to the hexaploid wheat line Vernstein from an Ethiopian durum (CI 7778) by Luig and Watson[38] was postulated as another *SR9* allele and named *Sr9e*[39]. Another resistance gene characterized from durum cv. Kronos in 2021, and temporarily designated as *SrKN*, was also located at the *SR9* locus and was postulated as *Sr9e*[40]. Loegering[41] then described a resistance gene in cv. Chinese Spring, *Sr9f*, was effective only against a widely avirulent *P. graminis* (*Pg*) isolate more characteristic of *Pg* f. sp. *secalis*. The near-isogenic line ISr9a-Ra carrying *Sr9a* in Chinese Spring background was susceptible to the same isolate providing evidence for allelism. *Sr9g* was identified in Thatcher[39] and probably was transferred to common wheat from durum cv. Iumillo. These authors identified the same gene in durum cultivars Acme and Kubanka. *Sr9h* identified in Gabo 56[30] was identical to genes described earlier in South African cv. Matlabas[42] and Canadian[34] cv. Webster RL6201.

Many of the *Sr9* alleles historically played important roles in stem rust resistance gene postulation and identifying *Pgt* pathotypes. *Sr9b* and *Sr9e* are the most commonly used resistance genes and are employed in all the routinely used international wheat stem rust differential sets in Australasia, North America, South Africa, India, and China[43,44]. The *Sr9a*, *Sr9d* and *Sr9g* have also been included in these differential sets and the latest defined BGRI international core differential set (Rusttracker.org).

Allelic series at a single locus are not uncommon and are major sources of genetic variation. There are 10 described alleles of the wheat powdery mildew resistance gene *PM3* (*Pm3a* to *Pm3j*)[45]. Similarly, the maize (*Zea mays*) common rust resistance locus *RP1* encodes complex alleles named from *Rp1-A* to *Rp1-H*[46]. The flax (*Linum usitatissimum*) *L* locus is another example with at least 12 resistance alleles, each conferring different arrays of response against flax rust caused by *Melampsora lini*[47]. Sequence polymorphisms among the *L* alleles in flax and *Pm3* alleles in wheat are considered to have arisen through point mutation and gene conversion events over time with the greatest variation in the LRR regions, consistent with their role in specificity

determination. Duplications and deletions in the LRR domains of some of the *L* alleles were observed and postulated to have resulted as a by-product of unequal crossing-over involving repeats in the LRR region[48]. Previous studies on *PM3*, *RPP13*, and *AVR-Pik* loci have also revealed these multiallelic resistance genes are thought to have evolved under strong diversifying selection to recognize specific *Avr* alleles from the pathogen[49–52].

The complexity of stem rust resistance loci with large allelic series has limited the development of allele-specific molecular markers, which are essential for pyramiding multiple *Sr* genes in breeding programs. The *SR9* locus on chromosome 2BL is a good example of this complexity comprising seven resistance alleles, *Sr9a*, *Sr9b*, *Sr9d*, *Sr9e*, *Sr9f*, *Sr9g*, and *Sr9h*[29,30,39,41,53,54]. Among these alleles, only *Sr9h* is effective against *Pgt* race TTKSK (or Ug99)[30]. In this study, we use gene cloning, complementation and comparative genetics to resolve the relationships among *Sr9* alleles and confirm their allelic identities.

## Results

### Identification of gene candidates for *Sr9b*, *Sr9e*, *Sr9g*, and *Sr9h*

Ethyl methanesulfonate (EMS) mutagenesis was undertaken on four lines which carry *Sr9b*, *Sr9e*, *Sr9g*, and *Sr9h*, respectively: (1) a Chinese Spring (CS) substitution line with a chromosome 2B pair from Kenya Farmer (KF) replacing its homologous pair (CS/KF 2B) (*Sr9b*), (2) cultivar Vernstein (*Sr9e*), (3) substitution line CS/Marquis (Mq) 2B (*Sr9g*), and (4) cultivar Gabo 56 (*Sr9h*). Approximately 1,000 spikes derived from 200–300 $M_1$ mutant plants of CS/KF 2B, Vernstein, and CS/Mq 2B were screened for response to Australian *Pgt* pathotypes 126-5,6,7,11 (Supplementary Fig. 1a), 21-0, and 343-1,2,3,4,5,6 Cook rust, respectively (Supplementary Data 1). Eight, seven and eight independent susceptible $M_3$ mutant lines, respectively, were confirmed from each mutagenized population by progeny testing. Similarly, eight independent mutant plants for *Sr9h* were obtained following infection of 1600 $M_2$ families from the EMS-mutagenized population of cv. Gabo 56 with *Sr9h*-avirulent Pgt pathotype TTKSK (Supplementary Fig. 1b). A 90 K wheat SNP analysis confirmed >97.7% marker identity between the Gabo 56 parent and mutant lines.

Mutational genomics and targeted exome capture of NLR immune receptor genes, a method termed MutRenSeq[14,55], was used for the isolation of *Sr9b*, *Sr9e_h1* (a specific *Sr9e* haplotype), *Sr9g*, and *Sr9h* from the above mutant lines. For each set of mutant plants, a common NLR encoding contig was identified for which most mutants varied by one SNP (Fig. 1a, Supplementary Figs. 2–5, Supplementary Data 2). In some instances, mutants apparently had arisen from deletions due to the absence of corresponding sequences in the mapping alignments. In the case of capturing the candidate gene *Sr9g* (Supplementary Fig. 4), only three mutants out of five were identified as carrying SNPs within the captured sequence region, this could possibly be because the nonsynonymous change may occur in regions other than the CDS of the target gene in the rest of the mutants (Supplementary Fig. 2). All the SNP mutations detected were transition events and resulted in nonsynonymous substitutions that caused either amino acid polymorphisms in the predicted protein sequence or premature stop codons (Fig. 1a, Supplementary Figs. 2–5). The candidate contigs for *Sr9b*, *Sr9e_h1*, *Sr9g*, and *Sr9h* were highly similar in sequence to each other, consistent with them being either alleles or related paralogous genes.

Each gene candidate was predicted to encode an NLR type R protein, protein sequence conservation among the five *SR9* alleles indicating the most polymorphic region is the LRR domain. The major polymorphisms are between the SR9B, SR9G, SR9H cluster and the two SR9E haplotypes (Supplementary Fig. 6). The predicted protein sequences of SR9B differed by a single amino acid from SR9G (M1169K) and SR9H (G1173R), respectively, whereas the SR9G and SR9H differed by two amino acid substitutions, K1169M and G1173R (Fig. 1b). AlphaFold2 prediction yielded a structural model containing

the expected NB domains as well as an extensive LRR containing 42 repeat units and formed a spiral with 1.25 turns. The two amino acid differences between SR9B, SR9G, and SR9H were located in the 26th LRR unit (Fig. 1b). The number of the LRR units of the SR9 allele encoded proteins was compared to other CNL type wheat Sr proteins based on the AlphaFold2 full length protein structure prediction. An unusual large number of 42 LRR units was observed for the SR9 locus and SR21, while the rest of the wheat SR have a majority LRR number of 17. The flax rust immune receptor L6 was included as an outgroup (Supplementary Data 3).

Each of the candidate contigs for *Sr9b*, *Sr9e_h1*, *Sr9g*, and *Sr9h* showed high sequence identity (97–99%) to *TraesCS2B03G1225900* (CS Reference genome RefSeq v2.1), which is part of a small family of an NLR cluster (seven genes, inclusive of pseudogenes) flanked by single copy genes *TraesCS2B03G1225800* and *TraesCS2B03G1227100*. Significantly, the genomic position of the single copy flanking gene, *TraesCS2B03G1225800*, enabled the identification of the first gene member of the NLR cluster (*TraesCS2B03G1225900*) as the lineage from which the *Sr9* candidate alleles were derived (Fig. 2, Supplementary Data 4). This position was further corroborated by the

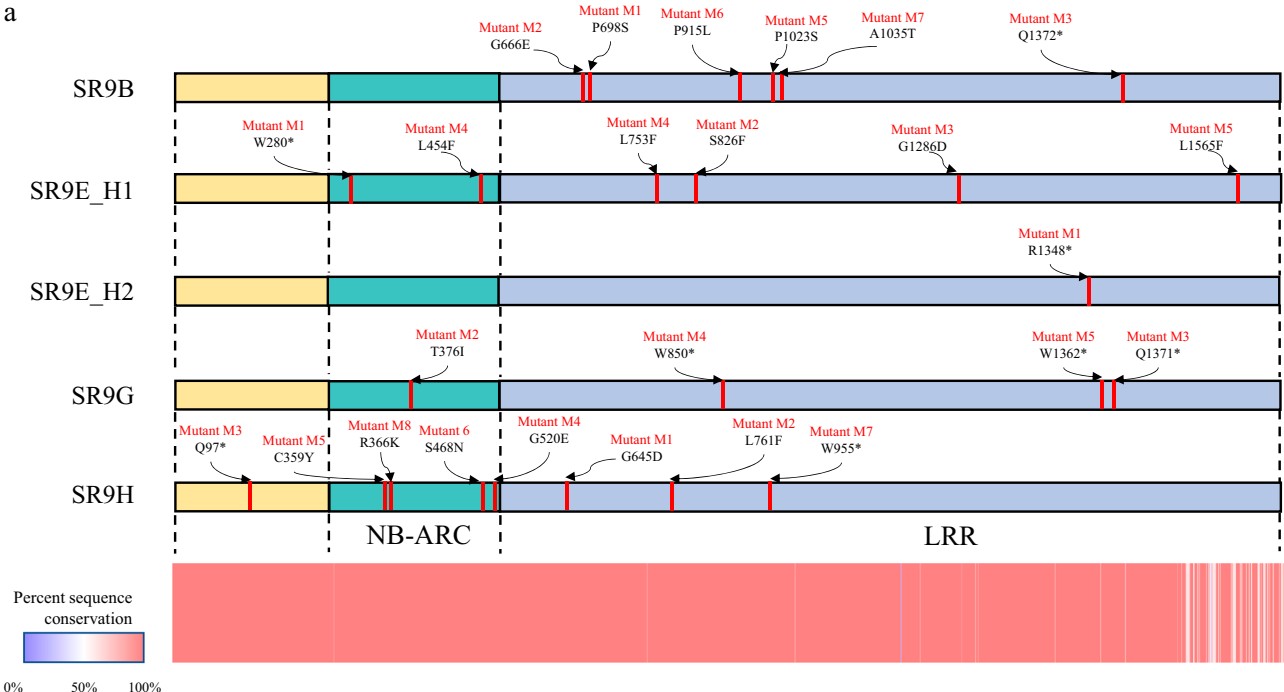

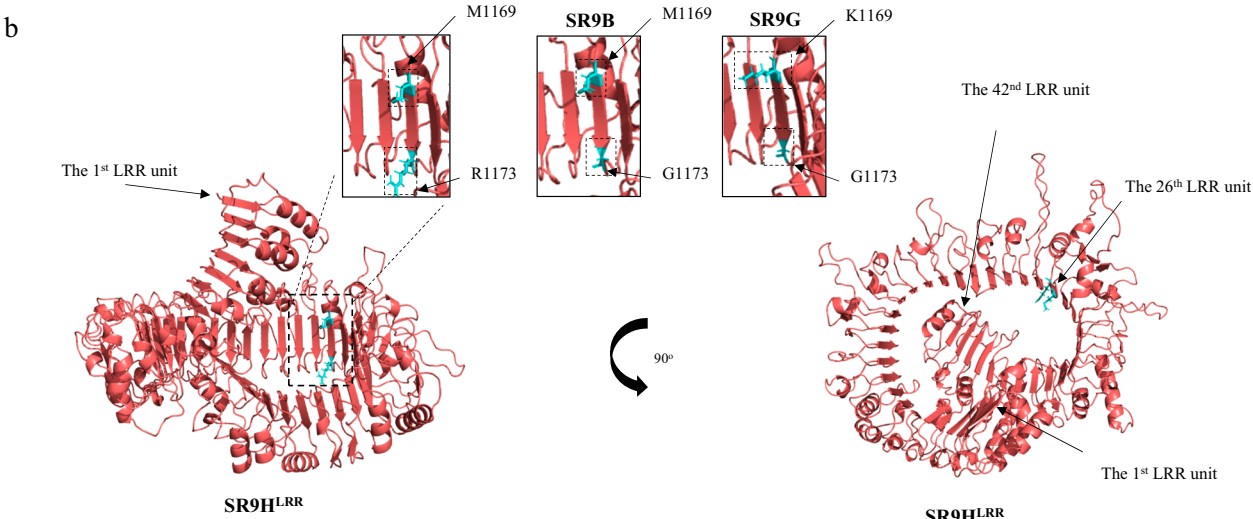

**Fig. 1 | Predicted protein composition of the *SR9* locus. a** The positions of mutations in *SR9* locus encoding proteins for SR9B, SR9E_H1, SR9E_H2, SR9G, and SR9H from mutant lines are shown in red, with predicted amino acid substitutions caused by missense mutations or premature stop codons shown above. N terminal domain, NB domain, and LRR motifs are shaded with yellow, cyan, and purple, respectively. Protein sequence conservation among the five *SR9* alleles is shown in the lower plot indicating the most polymorphic region is the C-terminal region of the LRR domain. **b** Structure of the LRR region of SR9H from AlphaFold2 prediction of full length protein highlighting the two polymorphic aa sites in the 26th LRR unit. Inset regions show close up of this region for SR9H, SR9B, and SR9G. The two polymorphic residues are shown in sticks and highlighted in cyan in the predicted surface-exposed horseshoe shaped beta strands.

publicly available wheat pan-genome of which cultivars CDC Landmark, Mace, SY Mattis and CDC Stanley shared identical sequences with *Sr9b* as well as a partial chromosome 2B sequence assembly of cv. Gabo 56, the source of *Sr9h*.

To further validate the *Sr9h* gene candidate, a KASP marker, *Sr9h6830*, was developed from the *Sr9h* contig identified by MutRenSeq (Supplementary Data 5) and its corresponding allelic variant *TraesCS2B03G1225900* in Chinese Spring. Mapping of this marker in Gabo 56/CS ($n = 104$)[30] mapping population showed co-segregation between the marker and *Pgt* race TTKSK resistance. Of a total of 104 plants from F$_{2:3}$ families, 35 plants were homozygous resistant (*Sr9h/Sr9h*), 48 plants were segregating (*Sr9h*/CS) and 21 plants were homozygous susceptible (CS/CS) ($\chi2$ 1:2:1 = 4.38, 2 df). The KASP marker genotypes were in perfect agreement with the *Sr9h* phenotypes. Similar results were obtained also from the CM664/CM670 population.

Mapping of *Sr9b* and *Sr9g* alleles in two respective F$_3$ populations, [CS*7/ KF 2B (*Sr9b*); $n = 99$] and [CS*7/Mq 2B (*Sr9g*); $n = 97$], showed co-segregation between the resistance and co-dominant, allele-specific markers derived from the candidate genes. In each family, resistance segregated as a single gene in the F$_2$ and F$_3$ populations. Out of 99 plants in the *Sr9b* (CS/KF 2B) population, 33 carried the CS allele (CS/CS), 48 were heterozygous (CS/*Sr9b*) and 18 carried the homozygous resistant allele (*Sr9b/Sr9b*) ($\chi2$ 1:2:1 = 4.64, 2 df). The 97 plants in the *Sr9g* population (CS/Mq 2B) consisted of 28 plants homozygous for the CS allele (CS/CS), 50 heterozygous (CS/*Sr9g*) and 19 homozygous for the *Sr9g* allele (*Sr9g/Sr9g*) ($\chi2$ 1:2:1 = 1.76, 2 df). The co-segregation of *Sr9b*, *Sr9g*, and *Sr9h* genotypes with resistance further supported the functionality of these detected resistance alleles.

Based on the high sequence similarity between candidate contigs of *Sr9b*, *Sr9e_h1*, *Sr9g*, and *Sr9h*, we undertook allele mining to identify a similar candidate for *SrKN* from durum cv. Kronos. This process was greatly simplified by the available partial genome sequences for Kronos.

## Two sources of *Sr9e* encode different haplotypes with similar resistance specificity

Both hexaploid Vernstein and tetraploid Vernal emmer have been used extensively as the resistance reference stocks carrying *Sr9e*. However, sequencing of the *Sr9e* homolog from Vernal emmer indicated that the latter contained a different gene sequence, encoding a protein with eight amino acid differences (Supplementary Data 6). The *SrKN* gene in Kronos was previously mapped in a large population (3366 gametes) from the cross Kronos × Rusty, to a cluster of NLR genes in a 5.6-Mb region in tetraploid wheat Svevo corresponding to a 7.2-Mb region in hexaploid wheat cv. CS. Among these NLR genes, *TRITD2Bv1G223210*, immediately adjacent to *TRITD2Bv1G223200* (corresponding to a single copy gene, *TraesCS2B03G1225800*), was identical to the *Sr9* sequence from Vernal emmer, consistent with the indistinguishable responses of Vernal emmer and *SrKN* to a range of *Pgt* races[40]. Based on these observations we conclude that *SrKN*, originally identified in Kronos, is the same *SR9* allele present in Vernal emmer.

Sequence analysis of the equivalent allele from an Ethiopian tetraploid (4x) accession (CI 7778), which was the source of *Sr9e* in Vernstein (6x), showed that *Sr9e*_CI 7778 has the same sequence as identified in Vernstein. We described the two variants or haplotypes as *Sr9e* haplotype 1 (*Sr9e_h1*), originally from CI 7778 (4x) and present in hexaploid Vernstein, and *Sr9e* haplotype 2 (*Sr9e_h2*) present in Kronos (4x) and Vernal emmer (4x).

To determine whether candidate gene *Sr9e_h2* (*TRITD2Bv1G223210*) confers resistance to *Sr9e*-avirulent *Pgt* race TRTTF (isolate 06YEM34-1), a mutant line T4-3163 with a premature stop codon in *Sr9e_h2* (R1348*) was selected from a sequenced EMS mutation population of Kronos. As Kronos carries stem rust resistance genes *Sr13a* and *Sr9e_h2*, we crossed mutant T4-3163 to Kronos *Sr13*

mutant line T4-3102, which carries a premature stop codon in the LRR domain of *Sr13a*[6]. Two F$_3$ siblings were selected, line 19059-27 homozygous for *Sr9e_h2* but lacking *Sr13a* (carrying *Sr13a* with a premature stop codon) and line 19059-26 lacking both *Sr9e_h2* and *Sr13a*. Based on seedling tests with race TRTTF (Sr9e_KN-avirulent)[40], line 19059-27 exhibited high levels of resistance to *Pgt* race TRTTF, whereas line 19059-26 was fully susceptible.

To determine the resistance profiles of *Sr9e_h1* and *Sr9e_h2*, we inoculated lines 19059-27 and Vernstein carrying each of these two haplotypes with 10 different *Pgt* races and found that they showed identical resistance profiles and similar resistance responses against all the tested races (Supplementary Fig. 1c, Supplementary Data 7).

## Transgenic complementation of *Sr9h* and *Sr9e_h2*

*Sr9h* and *Sr9e_h2* gene candidates were selected for validation experiments by transgenic complementation. The susceptible wheat cv. Fielder was transformed with an 8775-bp chimeric DNA fragment from cv. Gabo 56 encoding the complete *Sr9h* (5188 bp)-transcribed region and 2080 bp upstream and 1507 bp downstream regions from cv. Cadenza inserted before and after the start and stop codons, respectively. Stem rust response assays of T$_1$ transformants with *Sr9h*-avirulent *Pgt* race TTKSK (04KEN156/04) clearly established that the candidate of the *Sr9h* gene was sufficient to provide resistance in the Fielder background with significantly decreased pustule size compared to the Fielder control (Fig. 3a). Three of the five T$_1$ families segregated for TTKSK resistance, whereas in two families all plants were resistant. From each of the five transgenic *Sr9h* families, the average pustule sizes of infected leaves from plants that were PCR-confirmed to possess the transgene by *Sr9h* KASP marker *Sr9h6830* were compared to the 'Fielder' background. All five families showed significantly reduced pustule sizes compared to 'Fielder' ($p < 0.001$) (Supplementary Fig. 7). A total of six plants across three transgenic families did not possess the *Sr9h* transgene. We also compared the average pustule size of these progeny without *Sr9h* to Fielder and found no significant difference. We validated the effectiveness and race-specificity of *Sr9h* in transgenic families Sr9h-1a and *Sr9h*-5b (Supplementary Fig. 8). The two transgenic families showed significantly reduced average pustules sizes compared to Fielder for *Sr9h*-avirulent races TTKSK and RKRQC, but did not show significant differences from Fielder in response to Sr9h-virulent races QFCSC and MJGTC. Since the race-specificity of the *Sr9h* gene is confirmed in these experiments, we can conclude that the enhanced immunity of the transgenic lines is truly due to *Sr9h*-specific function, and not just a pleiotropic effect of multi-copy NLR expression in the transgenic lines.

For *Sr9e_h2*, a 10,974 bp genomic DNA fragment from durum cv. Kronos, including the complete coding region and introns (5197 bp) and 5' (3371 bp) and 3' (2406 bp) regulatory sequences, were transformed into Fielder. Fourteen transgenic T$_0$ plants contained the transgene as evidenced by being positive for PCR markers *pku4861F7R7* and *pku4861F4R4* that amplify regions of the SR9E_H2 LRR domain (Supplementary Data 5). When challenged with *Sr9e*-avirulent *Pgt* race 34MKGQM, these T$_0$ plants containing the transgene showed high levels of resistance along with the control introgression line, Fielder-*Sr9e_h2* (PI 700734) (Supplementary Fig. 9a). These plants produced *Sr9e_h2* transcripts that were significantly higher than the background homologs in the susceptible control Fielder ($p < 0.001$; Supplementary Fig. 9b). T$_1$ families consisting of approximately 20 plants from each transgenic event were inoculated with race 34MKGQM and the resistance was shown to co-segregate with the presence of the transgene (Fig. 3b).

Following the above validation of *Traescs2B03G1225900* homologs as the causal gene for *Sr9b*, *Sr9e_h1*, *Sr9e_h2*, *Sr9g*, and *Sr9h* alleles, we sought to identify the *Sr9a* and *Sr9d* alleles. In the absence of unambiguous mutants for *Sr9a* and *Sr9d*, we PCR amplified and sequenced the putative *SR9* alleles from Red Egyptian (RE, *Sr9a*) using

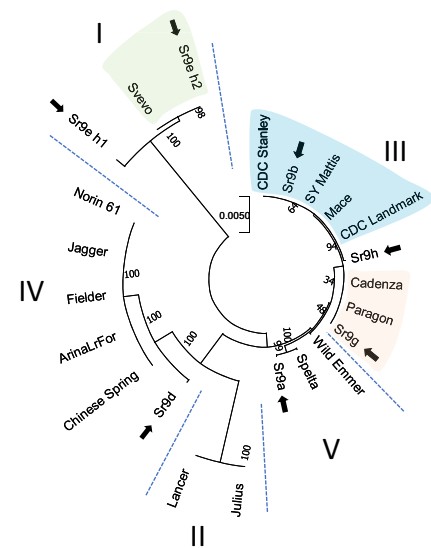

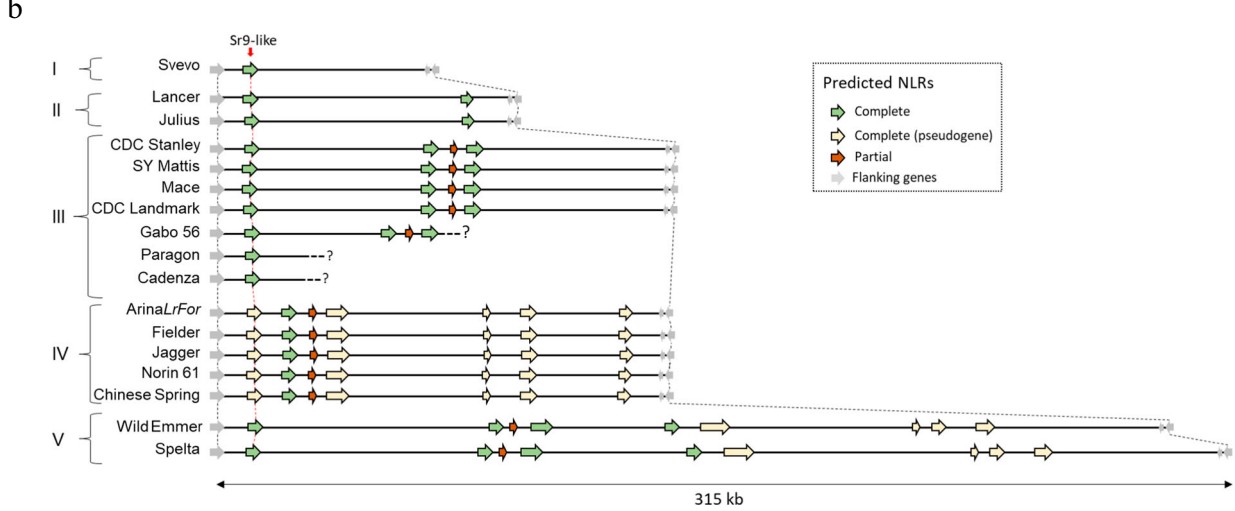

**Fig. 2 | NLR haplotype analysis of the wheat pangenome at the *SR9* locus on chromosome 2B. a** Phylogenetic grouping using genomic sequences of different functional *SR9* alleles (black arrow) and closest homologs in sequenced wheat genomes. Accessions carrying functional alleles have been highlighted. **b** Schematic of the predicted NLRs within the *SR9* homologous genomic region of wheat lines. NLRs are represented by colored arrows according to the NLR Annotator output as complete NLRs (green), complete pseudogenes (yellow) or partial NLRs (red). The position of closest *SR9* homolog (vertical red arrow connecting red dashed lines) is always the first NLR of the cluster. The numbering I to V highlights phylogenetic subgroups (**a**) which have a conserved structure in the NLR cluster (**b**).

chromosome substitution line CS/RE 2B and durum cv. Mindum and hexaploid cv. Renown (*Sr9d*). The inferred *Sr9a* and *Sr9d* candidate sequences with complete NLR open reading frames retained the same gene structure as the other validated *SR9* alleles with 98.8 and 99.7% sequence identity to *TraesCS2B03G1225900*, respectively. Sequence comparisons of the candidate SR9A protein with the validated *SR9* alleles, for example SR9B, showed amino acid substitution differences occurred predominantly in the LRR region, while SR9D differences were spread throughout the protein (Supplementary Fig. 6).

### *Sr9f* is atypical in the *SR9* allelic series

Analysis of the *SR9* locus in the Chinese Spring reference genome revealed that the candidate gene sequence, *TraesCS2B03G1225900* (we refer to *Sr9CS*), carried an in-frame stop codon (K258*) resulting in a truncated protein lacking the LRR and most of the nucleotide binding domain. *Sr9f* was first reported as an allele of *Sr9a* and present in CS (CI

14108). The gene was also reported in reference stock ISR9a-Sa (CI 14170), as a susceptible NIL corresponding to *Sr9a*, ISR9a-Ra (CI 14,169), developed in Chinese Spring background[41]. Unlike all the other *SR9* alleles, *Sr9f* is the only allele that requires low temperature (17 °C) to detect a resistance response. Additionally, the gene is unusual in conferring resistance to only a single reported *Pgt* culture, '111 × 36'. Previous studies reported low levels of temperature sensitivity in *Sr9a* and *Sr9b* where increased resistance was observed at higher temperatures[55], similar to *Sr21*[7,29,56,57]. Similarly, no difference in *Sr9e_h2* resistance was observed when tested at 18 °C day/15 °C night or 25 °C day/22 °C night (Supplementary Fig. 10).

To exclude the possibility that the Chinese Spring used for the reference genome differed from CI 14108, we recovered the original *Sr9a* and *Sr9f* stocks as the original *Sr9f*-avirulent *Pgt* culture '111 × 36' maintained at the USDA-ARS Cereal Disease Laboratory (St Paul, Minnesota) and confirmed their *SR9* allelic phenotypes (Supplementary

a

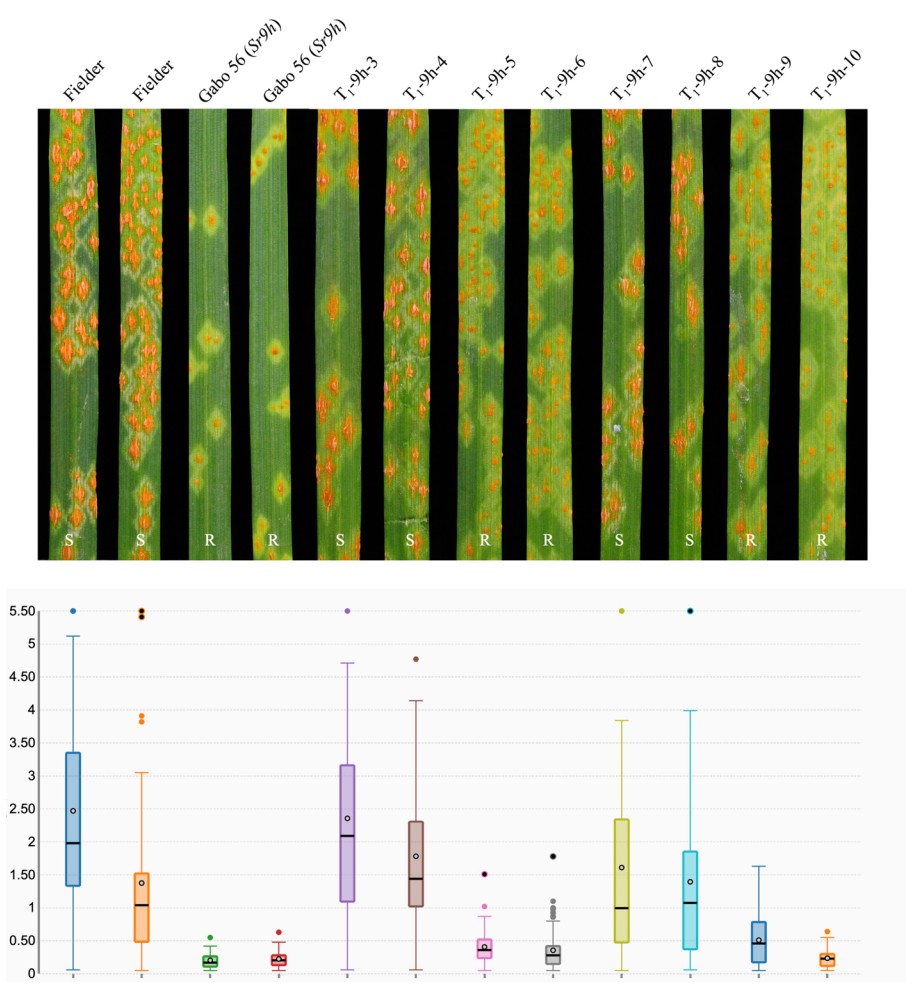

b

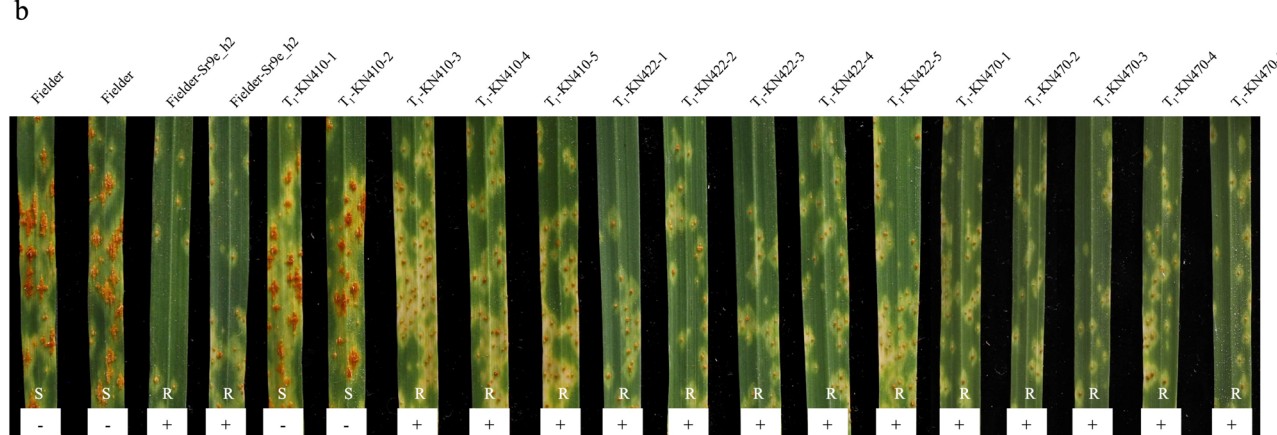

**Fig. 3 | Transgenic complementation of *Sr9h* and *Sr9e_h2*. a** *Sr9h* candidate confers resistance when transferred into the susceptible hexaploid wheat line Fielder. Seedling infection types of Gabo 56 (*Sr9h*), Fielder and eight plants from T₁ family No. 7 of *Sr9h* transformants in a Fielder background in response to *Pgt* race TTKSK (upper panel), along with the Sr9h-KASP marker assessed indicated by green labeling) and the average pustule size in mm² (indicated by black circled gray dot in each box) in the box and whisker plot (lower panel). The black center line denotes the median value, while the box contains the 25th to 75th percentiles of dataset. The whiskers mark the 5th and 95th percentiles. **b** Reactions to *Pgt* race 34MKGQM (isolate 20IAL06) in seven independent T₁ transgenic families. Each 5 plants in the figure representing approximately 20 plants from three transgenic families T₁-KN410, T₁-KN422, T₁-KN470 were inoculated and grown in a growth chamber at 25 °C day/22 °C night, along with the marker assessed ("+" indicates presence and "−" indicates absence, "R" for resistance and "S" for susceptible). Source data are provided as a Source Data file.

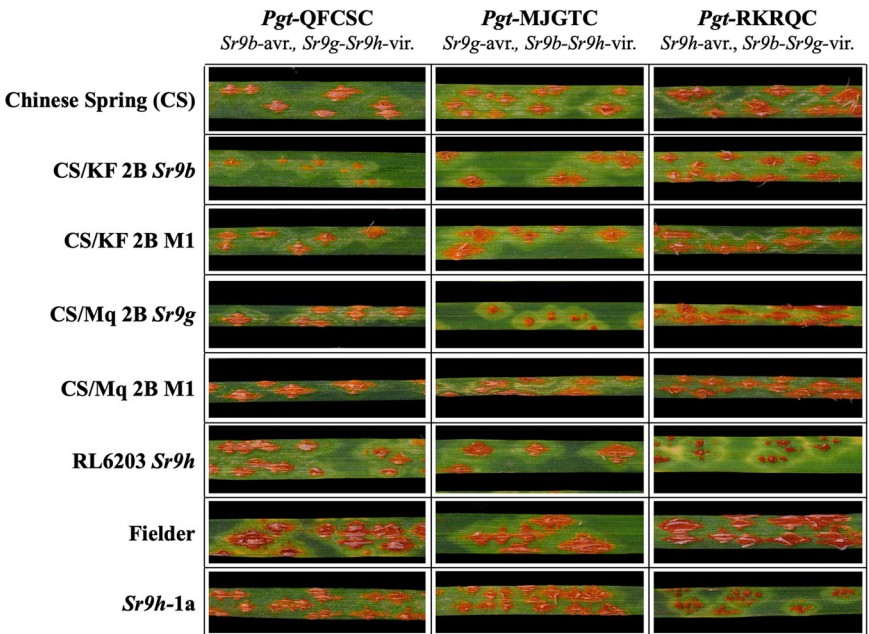

**Fig. 4 | Resistance race specificities of *Sr9b*, *Sr9g*, and *Sr9h*.** Images of selected leaves including genetic stocks without or with *Sr9b, Sr9g*, and *Sr9h* (CS/KF 2B, CS/Mq 2B, and RL6203, respectively) and *Sr9h* transgenic line Sr9h-1a in response to three races of *Pgt* (QFCSC, MJGTC, RKRQC) that show differential responses to *Sr9b*, *Sr9g*, and *Sr9h*.

Fig. 11). Resequencing of these reference stocks confirmed that CI 14108 and CI 14170 shared identical sequences with *Sr9CS*, whereas CI 14169 was identical to the putative *Sr9a* sequence. Since Loegering (1975)[41] showed that *Sr9f* was completely linked in repulsion with *Sr9a* and we found that the *SR9* homolog in the *Sr9f* genetic stock was likely not the functional *Sr9f*, it is therefore possible that a different member of the NLR cluster at the *SR9* locus or another closely linked gene in Chinese Spring conditions *Sr9f* functionality.

**Resistance race specificities of SR9B, SR9G, and SR9H**
After we validated the effectiveness and race-specificity of *Sr9h* in transgenic families, 12 different races of the wheat stem rust pathogen were inoculated onto monogenic lines possessing each of the *Sr9* alleles (*Sr9b*, *Sr9g* and *Sr9h*) to further demonstrate that each *Sr9* allele possesses a unique race specificity, in particular, to clearly explore the resistance profiles as they only differ by one or two amino acids (Supplementary Data 8). The *Sr9b*, *Sr9g*, and *Sr9h* genes were present in 'Chinese Spring' chromosome substitution line "CS/KF 2B", CS chromosome substitution line "CS/Mq 2B", and Fielder-background transgenic lines "*Sr9h-1a*" and "*Sr9h-5b*", respectively. In order to demonstrate the resistance changes among these three alleles, they were each compared independently to the corresponding mutant lines (for SR9B and SR9G) and non-transgenic 'Fielder' background (for SR9H). All materials, including additional controls 'Chinese Spring' and SR9H genetic stock 'RL6203' were inoculated with five stem rust pathogen races (Supplementary Data 9). Three of these races clearly discriminated the avirulence/virulence phenotypic effects of each allele: QFCSC (*Sr9b/Sr9g,h*), MGJTC (*Sr9g/Sr9b,h*), and RKRQC (*Sr9h/Sr9b,g*) (Fig. 4). For responses to these three races, we documented the average pustule sizes across leaves for each line-race combination and performed analyses of the mutants and transgenic lines compared to their respective controls (CS/KF 2B, CS/Mq 2B, and Fielder for SR9B, SR9G, and SR9H, respectively). The analyses clearly demonstrated unique race-specificity for each allele (Supplementary Figs. 8 and 12) where SR9B was only effective to race QFCSC, SR9G was only effective to race MGJTC, and SR9H was only effective to race RKRQC (in addition to TTKSK). These results show the differential effects of the residues

M1169 and R1173 between SR9B, SR9G, and SR9H as exhibited by race-specific resistance.

**Haplotype analysis of *Sr9e***
In addition to Kronos and Vernal emmer, previous studies postulated that several durum cultivars also carry *Sr9e*, such as Langdon[58,59], Lebsock[60], Svevo[40], and Leeds[61]. Sequencing of *Sr9e* amplicons in these durum cultivars confirmed the presence of genes 100% identical to *Sr9e_h2*. To further understand the sequence variation in *Sr9e*, we sequenced the coding regions of *Sr9e* from 80 *T. turgidum* subsp. *dicoccon* (4x) accessions using four pairs of gene-specific primers (Supplementary Data 5 and 10). The sequenced coding regions showed that none of them carried *Sr9e_h1*. Only two accessions, PI 94616 and PI94638, had a gene sequence that was 100% identical to the *Sr9e_h2* in Kronos. Line PI 94657 with a new haplotype differed from *Sr9e_h2* by only one amino acid (Supplementary Data 11). Although an array of stem rust responses for this line was not generated, it exhibited a high level of resistance to *Sr9e*-avirulent race 34MKGQM (Supplementary Fig. 13). Among susceptible accessions, four showed identical sequences and were classified as haplotype *sr9_h1*, differing from *Sr9e_h2* by eight amino acids (Supplementary Data 11). We also identified another five susceptible haplotypes (*sr9_h2* to *sr9_h6*) based on their susceptible reactions or/and shared amino acids (Supplementary Data 11). We failed to obtain PCR amplification products in some *T. dicoccon* accessions suggesting the presence of deletions or primer anchor region polymorphisms in many *T. dicoccon* accessions for which amplification of PCR products was unsuccessful. A comparison between the *Sr9e* susceptible and resistant haplotypes showed several critical polymorphisms. These results can assist in developing diagnostic markers for *Sr9e*.

**Conserved NLR clusters among allelic subgroups at the *SR9* and *SR21* loci**
The *Sr21* gene from *T. monococcum* present on chromosome 2A, a homoeolog of *Sr9*, is located in an NLR gene cluster which is structurally variable amongst wheat relatives. To gain a better understanding of the conservation and composition of the NLR cluster surrounding

the *SR9* locus on wheat chromosome 2B and its relationship with the homoeologous *SR21* locus, the corresponding intervals from the A, B and D subgenomes were examined from wheat and wheat relatives (Fig. 2, Supplementary Figs. 14 and 15). Two conserved non-NLR genes, *TraesCS2B03G1225800* and *TraesCS2B03G1227100*, flanking the immediate NLR cluster in Chinese Spring were used for homology search (BLASTn) against the pan-genome of wheat to delineate the loci. The position of the *Sr21* encoding sequence, as the first gene in the NLR cluster immediately flanking the homoeolog of *TraesCS2B03G1225800* in the diploid A subgenome, suggests that it is orthologous to the validated *Sr9* alleles in the B subgenome, and that the two are homoeologs (Supplementary Fig. 14).

The detected NLR clusters vary in length and complexity. The most complex cluster on chromosome 2B was found in *T. aestivum* ssp. *spelta* accession PI 190962, which comprises four full-length NLRs, four pseudogenes, and one partial NLR sequence. In contrast, Svevo only contains a single NLR at the locus (Fig. 2b, Supplementary Data 4). In comparison, *Sr21* only shared 99.13% similarity with its closest homologous sequence on chromosome 2A among available sequenced genomes. In addition, structural differences between the NLR clusters in the cloned *Sr21* donor line, DV92, and other reference genomes can be observed (Supplementary Fig. 14). Some accessions carry additional NLRs in front of the closest *Sr21* homologs, as previously reported in comparison to Chinese Spring and wild emmer accession Zavitan[7]. The overall complexity of the *Sr9* homologous region in chromosome 2D is greatly reduced. Only two different subgroups were found (Supplementary Fig. 15).

The *Sr9* alleles identified in this study can also be found in the pan-genome or amongst other partially sequenced wheat genotypes as stated earlier for *Sr9b*. Identical sequences to *Sr9g* were found in cultivars Cadenza and Paragon, consistent with previous findings of Cadenza as a carrier of *Sr9g*[62].

The Svevo genome shows that it has the *Sr9e_h2* allele as distinct from the *Sr9e_h1* allele in Vernstein. The *SR9-like* (*Sr9CS*) sequence variant in Chinese Spring is also present in Norin 61, Jagger, Fielder, and Arina*LrFor*. The putative *Sr9d* allele showed similarity to the truncated pseudogene-containing clade (Fig. 2a). The putative *Sr9a* sequence differs by two SNPs from the equivalent *T. aestivum* ssp. *spelta* PI 190962 sequence. Accessions within phylogenetic subgroups defined by the sequence of these *Sr9* alleles and *Sr9-like* NLRs showed strong structural conservation with comparable overall interval lengths and copy number of NLRs. Each of the *Sr9* alleles or highly conserved homologous sequences in other reference genomes was consistently the first gene in the NLR cluster immediately adjacent to the homolog of *TraesCS2B03G1225800*.

## Phylogenetic analysis of SR9H and SR21 with other NLR proteins

The five NLR proteins encoded by the *SR9* alleles *Sr9b*, *Sr9g*, *Sr9e_h1*, *Sr9e_h2*, and *Sr9h* were compared by phylogenetic analysis with other cloned NLR type SR proteins from wheat (Supplementary Fig. 16a) and NLR type proteins from other plant species (Supplementary Fig. 17a). SR9 proteins had the highest similarity to the stem rust resistance protein, SR21, with 85.14% and 86.02% identity to the SR9H and SR9E_H2 proteins, respectively. This SR9-SR21 clade is also clustered closely together with the BED domain-containing NLRs YR5a, YR5b, YR7, and falls quite apart from other NLRs with Coiled-coil domain at their N termini (Supplementary Fig. 17a). Interestingly, the SR21-SR9 proteins all appear to be lacking a predictable CC domain (Supplementary Figs. 17b and 18) and a highly conserved domain of 51 amino acids was identified towards the end of the N-terminal CC-equivalent domain of the SR9 and SR21 proteins that was not present in any other NLR proteins in the sequence alignment (Supplementary Fig. 16b). AlphaFold2 prediction located this 51 aa as the last α-helix of the N-terminal region of the full-length proteins of SR9 and SR21 that is

equivalent to the α4 helix in known CCs like ZAR1CC and SR35CC[20,24,25] (Supplementary Figs. 16c, 18, and 19).

Given the structural difference of the SR9 and SR21 N-terminal domains compared to other CC-NLRs, we tested whether these unique N-terminal fragments were auto active, as observed for other related CNL proteins[63]. C-terminal YFP tagged versions of the SR21 and SR9H CC-equivalent fragments (amino acids 1–216) including the additional sequence region were designed based upon their similarity with the corresponding auto active fragment of SR50 (SR50CC[1–163]) (Supplementary Fig. 16b). These chimeric proteins were transiently over-expressed in *Nicotiana benthamiana* and *N. tabacum* leaves via *Agrobacterium* infiltration and both SR21[1–216] and SR9H[1–216] triggered cell death in both *Nicotiana* species (Supplementary Fig. 16c).

## Discussion

In this study we identified wheat stem rust resistance genes *Sr9a*, *Sr9b*, *Sr9d*, *Sr9e_h1*, *Sr9e_h2*, *Sr9g*, and *Sr9h* and, as suggested by previous genetic analyses, confirmed that these genes are in fact an allelic series. The *Sr9* alleles in the B subgenome are encoded by a single NLR gene lineage and encode proteins that are homoeologs of the wheat stem rust resistance protein SR21 in the A subgenome, with protein sequence identities greater than 85%. Such an allelic series is unusual amongst resistance gene loci in wheat, with the only confirmed allelic series identified at the *SR13* locus and the powdery mildew resistance locus *PM3*[6,52,64].

In addition, we have shown that *Sr9e* and *SrKN*, with an identical *Pgt* resistance specificity based on 10 available *Pgt* races and therefore potentially the same gene, are in fact different haplotypes of the same gene (*Sr9e_h1* and *Sr9e_h2*). It is possible that the two haplotypes differ in their functions towards *Pgt* races that were not tested so far, or that one haplotype is able to recognize a newly emerging race in the future. Comparisons among the other *Sr9* alleles showed that in some instances, resistance specificity differences were determined by a single amino acid in the LRR region. Such a difference between *Sr9b* and *Sr9h* is responsible for different recognition specificities and the ability to recognize an avirulence gene in the Ug99 lineage (TTKSK).

Functional analyses demonstrated that the N-terminal CC-like domains from both SR21 and SR9H are able to trigger cell death when overexpressed in *N. benthamiana* and *N. tabacum*, despite containing an extra previously unreported 51-amino acid insertion relative to other CC domains. Previously a zinc finger BED domain was identified in the N-terminal regions of the YR5a/YR5b, YR7, and RPH15 resistance proteins[65,66]. These BED containing N-terminal domains were unable to trigger cell death in *N. benthamiana*[67], suggesting that additional components were required for signaling by these BED domain-containing NLRs. However, such a requirement does not appear to be needed for the SR21 and SR9H proteins as the 51-amino acid insertion-containing CC-equivalent domains are auto active in planta. This result could be potentially informative for studying the NLR type R protein structural conformation and signaling.

In summary, the *SR9* locus is an allelic series that encodes proteins closely related to SR21, which represents the homoeologous locus on the A genome of wheat. The close similarity between these homoeologs and their allelic series will enable further structural and functional analyses to determine the molecular basis of specificity. The flax rust *L* resistance locus, the maize common leaf rust *RP1* resistance locus, and the wheat powdery mildew *PM3* resistance locus, together with the identification and isolation of their corresponding avirulence (*Avr*) gene locus, proved to be a useful system to dissect host resistance specificity, pathogen pathogenicity, and *R-Avr* gene recognition/interaction and improve understanding of plant innate immunity. Unfortunately, we have limited information about R and Avr pairs in the wheat rust disease system. In comparison to the 16 wheat stem rust resistance genes cloned, only three corresponding *Avr* genes have

been isolated from the wheat rust pathogens, namely *AvrSr27*, *AvrSr35*, and *AvrSr50*[16,27,68]. The multiple *Sr9* alleles identified in this study will be useful tools to identify their corresponding *Avr* genes and investigate their resistance mechanisms. A better understanding of the evolution of disease resistance allelic series can generate valuable information on the durability of different mutations and help in engineering a more sustainable broad-spectrum resistance to the devastating rust pathogens.

## Methods

### Plant materials, mutagenesis, and mutant DNA extraction
Wheat lines carrying *Sr9a*, *Sr9b*, *Sr9d*, *Sr9e*, *Sr9g*, and *Sr9h* were mutagenized with EMS and progeny susceptible to corresponding avirulent *Pgt* races (listed in Supplementary Data 1) were selected[69]. Genomic DNA was extracted from seedling leaves[70]. DNA quality and quantity were assessed with a NanoDrop spectrophotometer (Thermo Fisher Scientific, Waltham, MA, USA) and by electrophoresis in 0.8% agarose gels. The sequenced EMS mutagenized population of the tetraploid wheat variety Kronos is available online (http://www.wheat-tilling.com/)[71].

### R gene enrichment and sequencing
Enrichment of NLRs was undertaken by Arbor Biosciences (Ann Arbor, MI, USA) using the myBaits protocol. The Triticeae NLR bait libraries Tv3 (*Sr9b*, *Sr9e*, and *Sr9g*) and Tv2 (*Sr9h*), are available at Github [https://github.com/steuernb/MutantHunter/blob/master/Triticea_RenSeq_Baits_V3.fasta.gz] and [https://github.com/steuernb/MutantHunter/blob/master/Triticea_RenSeq_Baits_V2.fasta], respectively. Library construction was undertaken using the TruSeq RNA protocol v2. All enriched libraries were sequenced using a HiSeq 2500 (Illumina) sequencing platform that generated 250 bp paired-end reads.

### Sequence analysis
The *Sr9h* candidate was identified with the MutRenSeq pipeline[72]. CLC Genomics Workbench v10.0 (Qiagen, Hilden, Germany) was used (*Sr9b*, *Sr9e*, and *Sr9g*) for reads quality control, trimming, and de novo assembly of wild-type reads using the following parameters: minimum contig length: 250, auto-detect paired distances, perform scaffolding, mismatch cost: 2, insertion cost: 3, deletion cost: 3, length fraction 0.9, and similarity fraction 0.9–0.98. For mapping sequence reads from wild-type and mutants against the de novo wild-type assembly the following parameters were used: no masking, linear gap cost, length fraction 0.5–0.9, and similarity fraction 0.95–0.98. Contigs containing mutations in each mutant line of *Sr9b*, *Sr9e*, and *Sr9g* and gene candidates were identified using the MuTrigo Python package (https://github.com/TC-Hewitt/MuTrigo) for SNP calling with default parameters.

### Gene sequence confirmation
Total RNA was extracted using a PureLink™ RNA Mini Kit (Invitrogen, Carlsbad, CA, USA) according to the manufacturer's instructions. cDNA synthesis was performed as described by Clontech (Mountain View, CA, USA). Nonsynonymous substitutions of *Sr9b*, *Sr9e*, and *Sr9g* identified in mutants by RenSeq were confirmed by PCR amplification of mutant DNAs and Sanger sequencing. Exon–intron structures of *Sr9b*, *Sr9e*, and *Sr9g* were confirmed by cDNA amplification and sequencing.

### Candidate gene confirmation by wheat transformation
The *Sr9h* gene candidate was introduced into wheat cultivar Fielder by *Agrobacterium*-mediated transformation using binary vector pVec-BARII and phosphinothricin as a selective agent[73]. $T_0$ plants were transplanted to a growth cabinet (23 °C, 16 h light/8 h darkness) and seeds of the $T_1$ family were harvested. $T_1$ families were inoculated with

*Sr9h*-avirulent *Pgt* race TTKSK (isolate 04KEN156/04). Rust-infected leaves were imaged and assessed 10–15 days post-inoculation. Images were analyzed and average pustule size was determined for each leaf by ASSESS 2.0 (2008)[6]. For *Sr9e_h2*, we used the PrimeStar Max DNA Polymerase (TaKaRa, Tokyo, Japan) to amplify a 10,974-bp genomic DNA fragment from the durum wheat cv. Kronos by PCR. This fragment was recombined into the linearized binary vector pCAMBIA1300 using the In-Fusion® HD Cloning Kit (Clontech) following the manufacturer's instruction. The resulting plasmid was transformed into cv. Fielder using *Agrobacterium tumefaciens* strain EHA105-mediated transformation[73] in the transformation facility, Peking University Institute of Advanced Agricultural Sciences. $T_0$ and $T_1$ transgenic plants were inoculated with *Sr9e*-avirulent *Pgt* race 34MKGQM (isolate 20IAL06). qRT-PCR primers SrKN-RTF1R1 (Supplementary Data 5) were used to estimate *Sr9e_h2* transcript levels in $T_0$ transgenic plants using *ACTIN* as the endogenous control. Transcript levels were presented as fold-*ACTIN* levels using the $2^{-\Delta Ct}$ method[74,75].

### Stem rust phenotyping
Stem rust responses of lines carrying *Sr9b*, *Sr9e*, and *Sr9g* together with corresponding mutants were determined on infected seedlings in a greenhouse[69], *Pgt* races were specifically chosen for avirulence to each gene. Stem rust phenotyping of seedling responses to *Sr9f* and *Sr9h* were conducted at 18 ± 2 °C with a photoperiod of 16 h[76] except that tests with *Sr9f* were conducted in a growth chamber maintained at 17 °C.

### Phylogenetic analysis
R protein sequences from the NCBI database were aligned using Expresso (https://tcoffee.crg.eu/apps/tcoffee/do:expresso) and phylogenetic trees in Supplementary Figs. 16 and 17 were constructed using CLC Sequence Viewer v8.0 using the neighbor-joining method with 1000 bootstrapping. The tree is shown in a circular phylogram format. Phylogenetic trees in Fig. 2, Supplementary Figs. 14 and 15 were constructed using Mega 10 (https://www.megasoftware.net)[77]. Genomic sequences (ATG to TGA) were aligned using Muscle algorithm using default settings. A maximum-likelihood tree was constructed using Tamura 3-parameter model and Gamma distribution (T92 + G) and test of phylogeny with 1000 bootstrapping.

### NLR haplotype analysis
The genes flanking the immediate cluster in the Chinese Spring reference sequence were used as query for similarity search (BLASTn). CDS sequence of the left flanking gene *TraesCS2B03G1225800* and right flanking query gene *TraesCS2B03G1227100* provided unique BLAST hits for chromosomes 2A, 2B and 2D with a similarity above 90%. The resulting coordinates from BLAST analysis against complete pseudomolecules of wheat[78] were used to extract respective sequence intervals with SAMtools version 1.9.0[79]. The NLR prediction and annotation were performed by NLR-Annotator (https://github.com/steuernb/NLR-Annotator) using default parameters[80].

### Protein structure predictions and analyses
The CC domain prediction was done by DeepCoil (toolkit.tubingen.mpg.de)[81]. The sequences of the proteins listed in Fig. 1b, as well as Supplementary Figs. 16c, 18, and 19 were structurally modeled using AlphaFold 2.0[82]. Full database was downloaded on 14/09/2021. The program was run with parameters "max_template_date=2021-10-06" and "preset=full_dbs". Structural comparison and analysis were carried out and produced by Pymol V2.2.0 (https://pymol.org/2/).

### Construct generation, protein extraction and immunoblotting
All selected CC domains were cloned into pDONR vector without stop codons and transferred into destination vectors pBIN19 with

C-terminal YFP fusions by Gateway cloning (Invitrogen, USA). Sequences were checked after each transformation. Protein was extracted from *N. benthamiana* leaves[16]. For immunoblotting analysis, proteins were separated by SDS-PAGE and transferred to a nitrocellulose membrane. Membranes were blocked in 5% skimmed milk and probed with anti-GFP (Roche, Germany). Labeling was detected using the SuperSignal West Femto Chemiluminescence kit (Thermo Scientific, USA). Membranes were stained with Ponceau S to confirm equal loading.

**Plant growth conditions and transient expression analyses**
*N. benthamiana* and *N. tabacum* plants were grown in a growth chamber at 23 °C with a 16 h light period. For transient expression analyses in *N. benthamiana*, pBIN19-derived vector constructs were transformed into *A. tumefaciens* strain GV3101. Bacterial strain was grown in LB liquid medium containing appropriate antibiotics at 28 °C for 24 h. Bacteria were harvested by centrifugation, resuspended in infiltration medium [10 mM MES (pH 5.6), 10 mM MgCl$_2$, and 100 μM acetosyringone] to an OD600 ranging from 0.5 to 1, and incubated for 2 h at room temperature before leaf infiltration. For every independent experiment, each construct was infiltrated on three leaves from three or four individual plants. For documentation of cell death, leaves were photographed 2–5 days after infiltration.

**Reporting summary**
Further information on research design is available in the Nature Portfolio Reporting Summary linked to this article.

## Data availability
The datasets and plant materials generated and analyzed during the current study are available from the corresponding authors. The data that support the findings of this study are openly available in NCBI, annotated genomic sequences of *Sr9b*, *Sr9g*, *Sr9e_h1*, *Sr9e_h2*, and *Sr9h* have been deposited at NCBI GenBank with accession numbers OP219803 (*Sr9b*), OP219804 (*Sr9g*), OP219805 (*Sr9e_h1*), OP219806 (*Sr9e_h2*), and OP219802 (*Sr9h*). Source data are provided with this paper.

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

## Acknowledgements

J.Z. acknowledges the support from CSIRO CERC Postdoctoral fellowship. S.C. acknowledges support from the Provincial Natural Science Foundation of Shandong (ZR2021MC056 and ZR2021ZD30) and the Open Project Funding of the State Key Laboratory of Crop Stress Adaptation and Improvement. B.B.H.W. acknowledges the Biotechnology and Biological Sciences Research Council (BBSRC) Designing Future Wheat Cross-Institute Strategic Programme (BBS/E/J/000PR9780). Jorge D. acknowledges support from the Howard Hughes Medical Institute (https://www.hhmi.org/) and competitive Grant 2022-68013-36439 (WheatCAP) from the United States Department of Agriculture, National Institute of Food and Agriculture. P.Z., R.Mc., and S.H. acknowledge the support from Grains Research and Development Corporation (GRDC), Australia. M.N.R. acknowledges support from the Durable Rust Resistance in Wheat project funded by the Gates Foundation and UK DFID, a fellowship under the OECD Co-operative Research Programme: Biological Resource Management for Sustainable Agricultural Systems, and the USDA-ARS National Plant Disease Recovery System.

## Author contributions

P.Z., M.N.R., S.C., R.Mc., C.H., Jorge D. and E.L. designed research; J.Z., J.N., S.C., M.J., M.K., T.H., H.L., E.E., K.S., S.H., D.B., R. A-K., C.H., R.Mc., P.Z., M.N.R. and E.L. performed research; J.Z., J.N., S.C., M.J., B.S., T.H., P.D., Jaroslav D., B.B.H.W., M.A., R.Mc., Jorge D., E.L., P.Z. and M.N.R analyzed data; J.Z., J.N., S.C., M.J., M.A., R.Mc., P.Z., E.L. and M.N.R. wrote the paper. J.Z., J.N., S.C., and M.J. contributed equally.

## Competing interests

The authors declare no competing interests.

## Additional information

¹CSIRO Agriculture & Food, Canberra, ACT 2601, Australia. ²Plant Breeding Institute, School of Life and Environmental Sciences, University of Sydney, Cobbitty, NSW 2570, Australia. ³State Key Laboratory of Wheat and Maize Crop Science, National Wheat Innovation Centre, Centre for Crop Genome Engineering, and College of Agronomy, Longzi Lake Campus, Henan Agricultural University, Zhengzhou 450046, China. ⁴Department of Plant Pathology, University of Minnesota, St. Paul, MN 55108, USA. ⁵Peking University Institute of Advanced Agricultural Sciences, Shandong Laboratory of Advanced Agricultural Sciences at Weifang, Weifang, Shandong 261000, China. ⁶John Innes Centre, Norwich Research Park, Norwich NR4 7UH, UK. ⁷Institute of Experimental Botany of the Czech Academy of Sciences, Centre of the Region Haná for Biotechnological and Agricultural Research, 77900 Olomouc, Czech Republic. ⁸US Department of Agriculture-Agricultural Research Service, Cereal Disease Laboratory, St. Paul, MN 55108, USA. ⁹School of Molecular Biosciences, College of Medical, Veterinary and Life Sciences, University of Glasgow, Glasgow G12 8QQ, UK. ¹⁰Plant Science Program, Biological and Environmental Science and Engineering Division, King Abdullah University of Science and Technology (KAUST), Thuwal 23955-6900, Saudi Arabia. ¹¹Centre for Desert Agriculture, KAUST, Thuwal 23955-6900, Saudi Arabia. ¹²Agriculture and Agri-Food Canada, Morden Research and Development Centre, 101 Route 100, Morden, MB R6M 1Y5, Canada. ¹³Department of Plant Sciences, University of California, Davis, CA 95616, USA. ¹⁴Howard Hughes Medical Institute, Chevy Chase, MD 20815, USA. ¹⁵These authors contributed equally: Jianping Zhang, Jayaveeramuthu Nirmala, Shisheng Chen, Matthias Jost. ✉e-mail: peng.zhang@sydney.edu.au; Matthew.rouse@usda.gov; evans.lagudah@csiro.au

