## [Peer Review File · Nature Communications]

Single amino acid change alters specificity of the multi-allelic wheat stem rust resistance locus SR9Reviewers' Comments:

Reviewer #1:

Remarks to the Author:

Zhang et al reported the isolation of five alleles at wheat stem rust resistance gene locus Sr9 located on chromosome 2B, namely Sr9b, Sr9g, Sr9e_h1, Sr9e_h2 and Sr9h, proved that the previously reported genes Sr9e and SrKN were actually two different haplotypes of Sr9e (Sr9e_h1 and Sr9e_h2). The authors also found these alleles encode a highly related nucleotide-binding site leucine-rich repeat (NB-LRR) type immune receptor, with an unreported N-terminal motif and an unusual long LRR domain. The protein encoded by the allele gene Sr21 from *T. monococcum* present on chromosome 2A also has the same characteristics like SR9 protein above, due to the close similarity between A and B subgenome. An interesting phenomenon that deserves further structural and functional exploration is that the two amino acid differences between SR9B, SR9G and SR9H, located in 26th LRR affects the resistance spectrum of wheat stem rust pathogen. This paper provides a better understanding of the evolution of wheat disease resistance allelic series.

The paper is in-depth in genomics research, but lacks further corresponding functional exploration. My recommendation would be to publish this paper after my specific comments are addressed.

Major issues:

1. Page 2 line 45: The sentence "that confers resistance to a unique spectrum of isolates of the wheat stem rust pathogen". I think you should do some experiments to compare the spectrum against wheat stem rust pathogen between this NLR and common NLR.
2. SR9H differs from SR9B and SR9G by only one and two residues, respectively, however only SR9H is resistant to Ug99. Here, it is necessary to substitute the residues M1169 and R1173 between SR9H, SR9B and SR9G, exchange the basic amino acids (arginine and lysine), and explore the resistance changes. In addition, it is not enough to show the polymorphic residues in LRR model, please analyze the structure and perform functional verification.
3. The prediction results of DeepCoil do not fully demonstrate the absence of CC domain in SR9 proteins, the model of SR21 and SR9H N-terminal domain looks similar to the structure of autoinhibited ZAR1 CC domain, would you provide a structure superposition and other stronger evidence. In addition, the SR50CC1-163-YFP and YFP proteins expressed in tobacco should also be detected by the immunoblot, and it would be better if the samples in Western blot could be on the one gel.

Minor issues:

1. Page 3 line 71: "calcium channel" is inaccurate, ZAR1 resistosome is a calcium-permeable cation channel.
2. Page 3 line 72: "CryoEM" should be written as "Cryo-EM".
3. Page 3 line 73 and 75: delete the "-" in "NLR – binds" and "Anderson-with".
4. Page 5 line 138: "343-1,2,3,4,5,6, respectively (Table 1)", I can't find Table 1 in the manuscript, maybe it is Table S1.
5. Page 7 line 219: I didn't find Figure S5C in paper.
6. Page 12 line 382: "N terminal" should be written as "N-terminal".
7. Please unify the following two formats, page 14 line 452: "10-15 days" and page 16 line 517: "2-5

d”

8. Reference 8, 9, 12, 19, 20, 26 and 65: The citation formation about “Science (80-.)” is incorrect, please revise and check the rest of reference.
9. Reference 16, 21, 22, 29, 60, 64 72, 74: Reference information is incomplete, please supplement and check all.
10. Figure 2 and S7 legend last sentence: “The numbering I to IV highlights phylogenetic subgroups...” maybe it should be “The numbering I to V highlights phylogenetic subgroups...”.
11. Figure S8a: The numbering I and II can be marked on the periphery of the phylogenetic grouping as in Figure 2 and S8.
12. Table S3: The primer CT21CC_attb and CT9hCC_attb, change all lowercase letters to uppercase.
13. Table S8: The footnote with the meaning of different colors in the table will be useful.

Reviewer #2:

Remarks to the Author:

Review of Zhang et al NCOMMS-22-43739, Nov-2022

A single amino acid change can alter the specificity of the multi-allelic wheat stem rust resistance locus SR9

The authors open their study of the SR9 locus with a review of the fifteen-wheat stem rust resistance genes that have thus far been cloned. They point out that eleven of the fifteen encode a single class of protein, namely nucleotide-binding leucine-rich repeat (NLR) immune receptor proteins. The authors then launch into a brief description and characterization, including the discovery of each of the eight alleles, a thru h, that made up the original SR9 locus. As pointed out by the authors, Sr9c was dropped and later renamed Sr36 (BGRI). This, plus the more recent addition of Sr9h, left us with the seven Sr9 alleles we know today.

As noted by the authors, the seven alleles of the SR9 locus have been deployed in agriculture to varying degrees. Alleles Sr9b, e, and g have been especially present in wheat cultivars. The most recently discovered Sr9h may also have a major impact because it is the only Sr9 allele that confers resistance to race TTKSK, or Ug99, and thus it has the potential to become an important wheat gene. The introduction has done a good job of describing the problem, briefly introducing the alleles, and making the reader aware of their importance.

KEY RESULTS: Procedures are similar to Zhang et al (2017). The alleles of the SR9 locus were sequenced and in doing so, there was more than one important fact established. First, six of the seven putative alleles were confirmed by molecular methods to be allelic, the exception being Sr9f whose status is left uncertain. The Sr9e allele was found to have two haplotypes, and possibly a third, which have thus far not been differentiated into separate alleles. Multiple haplotypes were not reported for any other allele. An interesting fact was that NLR clusters of these alleles are highly conserved, differing from each other by only one or two base-pairs. The conserved nature of the NLR clusters also allowed the authors to make perhaps the most unique conclusion of the study. They found when comparing the NLR clusters of the SR9 loci to that of the other fifteen cloned stem rust genes, that the Sr9 alleles and Sr21 are orthologous. While there have been many orthologous series of genes noted in wheat, to my knowledge, this is the first orthologous series established for stem rust resistance genes in wheat. The authors close the results section on the hope that publication of these gene sequences will be useful for studies of R-Avr (resistance-avirulence) interactions in plants.

VALIDITY: I don't find any flaws in the manuscript.

ORIGINALITY AND SIGNIFICANCE: The conclusions are definitively original and of immediate interest to people in my discipline and should be of interest to people in several other disciplines including

plant pathologists, plant breeders, plant geneticists, general botanists, and general geneticists.
DATA AND METHODOLOGY: As noted earlier, the approach that the authors took with this study is similar to the approach of Zhang et al (2017), a study in which Rouse was also one of the lead scientists. I think the similarities in methodology and data presentation is fine. The reported methods are sufficiently detailed to enable reproduction of the results. Accession numbers are reported to enable testing of lines.

APPROPRIATE USE OF STATISTICS AND TREATMENT OF UNCERTAINTIES: I have no problem with the general use of statistics in the manuscript. However, I do have some concerns with the use of the Stakman scale (Stakman et al, 1962) for reading rust. Many readers will be unfamiliar with this scale. I am familiar with the scale so I can easily follow what the authors are trying to convey. But additional detail may be needed for readers unfamiliar with Stakman. Specifically, I think it would be useful for each plant to be marked as R or S. This has been done for some Figures, but not all. Take Figure 1a as an example. The authors mention in the legend that there is a mixture of R and S sibling plants in Figure 1A. Based on the numbering and infection types, it is easy enough to determine which are the sibling plants. But perhaps not so to an untrained eye, so identifying plants as R or S would be useful. In discussing uncertainties and the Stakman scale, one Figure comes to mind, Figure S6. Based on the haplotypes shown in the legends, the two leaves to the left should be resistant, and they clearly are. The remaining 8 leaves should be susceptible, and seven are clearly susceptible, but the infection type shown for PI94674 is not in my opinion a clear susceptible phenotype. Do the authors have any comments? Perhaps this is another case where identifying plants as R or S would be useful.

CONCLUSIONS: Yes, I find the conclusions robust, valid, and reliable.

SUGGESTED IMPROVEMENTS: One item that I looked for in the manuscript was a comparison of the differences in the Sr9 alleles. The authors report that there is only a one or two base pair difference between these alleles, but these allelic differences are not listed in a table. The authors do report at the end of the manuscript that the allele sequences are available at the NCBI GenBank and they report the accession numbers. But it would be nice if a summary could be presented in a brief supplemental table. Perhaps there is a reason not apparent to me that this was not done. Basically, what I am arguing is expanding Table S4 to include all alleles.

EDITS: On line 145, "MutRenSeq1451" should be changed to "MutRenSeq14, 51"

REFERENCES: All okay.

CLARITY AND CONTEXT: All okay

BGRI 2022. <https://globalrust.org/gene/sr36>

Stakman, E.C., Stewart, D.M. & Loegering, W.Q. (1962) Identification of physiologic races of *Puccinia graminis* var. *tritici*. USDA ARS E-617. US Government Printing Office, Washington, DC.

Zhang, W. et al. (2017) Identification and characterization of Sr13, a tetraploid wheat gene that confers resistance to the Ug99 stem rust race group. *Proc Natl Acad Sci U S A* 114, E9483–E9492.

Reviewer #3:

Remarks to the Author:

This paper describes the molecular identification of an allelic series of resistance genes against the wheat stem rust pathogen in wheat. The Sr9 alleles were identified and characterized using mutational analysis, genetic complementation by transformation and phylogenetic analysis based on genomic resources. The work reveals a series of genes that are very similar in sequence (some of them with only one or two amino acid polymorphisms), yet each of them providing a characteristic resistance spectrum to stem rust isolates. This makes the work highly valuable for understanding the molecular basis of specificity, although the recognized proteins (avirulence proteins) in the pathogen remain to be identified. There are currently no allelic series of resistance genes to rust known in the wheat genome, and there is only one well characterized allelic series against the mildew pathogen. This makes the study an original and novel contribution to understanding diversity of resistance genes in the globally important wheat crop. The work also identifies the Sr9 alleles and the Sr21 gene on the A

genome as homoeologous genes, adding an important piece of data of relevance for the study of resistance gene evolution.

There are some shortcomings in the manuscript which mostly relate to data presentation, content of presented data and the depth of data analysis. Most importantly, there is a lack of clarity which makes the evaluation of the work difficult.

1. The abstract mentions that Sr9a was characterized in this study. However, there are no data on Sr9a in the manuscript and it is unclear from the description of the work what is the reason of this.
2. Figure 1a is unclear and needs a sister figure with sequence comparisons (aa sequence polymorphisms distinguishing the alleles; in addition to mutant sites given in the figure). The few aa polymorphisms that distinguish the different alleles (again: where is Sr9a?) must be clearly indicated so the reader gets an overview on the diversity, also related to the domain structure.
3. Where is Sr9d in Figure 1a?
4. At the bottom of Figure 1a there is a scheme that shows the diversity in the allelic sequences (down to 50% in the most C-terminal region?). This diversity is much higher than it is described in the text. Where do all the polymorphisms at the C-terminal end come from? Does this include paralogs, homoeologs, or Sr9a? This needs more clarity and a more detailed description in the text.
5. Figure 2 legend: Different SR9 alleles are shown. However, it is unclear what are functional alleles, what are paralogs. There is lack of clarity. It remains unclear how sequences that differ only in very few aa can give such differences in phylogenetic groupings. Or is this all relating to polymorphisms present in Sr9d and Sr9e (but their diversity is not addressed in the manuscript)?
6. The aa polymorphisms between the different allelic variants must be described in much more detail (see above comment 2). In the current version of the manuscript, it is not possible to fully understand the work based on the given information.
7. What is the unreported motif at their N termini mentioned in the abstract? This is not described any further in the manuscript.
8. There are very interesting observations on temperature sensitivity of different alleles. But this is not analyzed in depth, and it is unclear how this contributes to the study. The authors could study temperature dependence of autoactive versions of the allelic variants in transient expression assays in *Nicotiana* to verify that temperature differences are really caused by the NLRs themselves and do not relate to genetic background effects.
9. It remains unclear how the experimental findings on the N-terminal domain contribute to the study. What are the conclusions of Figure S9 and Figure 4, and what is the purpose of these experiments?

Other comments:

Figure 1a: please indicate aa positions of the borders of the domains

Figure 1 legend: ... nonsense mutations or premature stop codons... : These terms mean the same. It should be missense or premature stop codons.

Supplementary figures:

Figure S2 contains information in unreadable small fonts. The figure must be edited and improved for clarity.

Figure S7b: The figure needs improvement for clarity. Sr21 was identified in DV92. Please indicate clearly which of the genes indicated in the figure of the DV92 genome is Sr21.

Related to Figure S7: Is anything known if the inversion of haplotype 2 has resulted in a known suppression of recombination at this locus in breeding programs?

Line 202: which Sr9 sequence?

There are few language problems in the manuscript and some editing is necessary: e.g.

L65: ... and most of them have a predictable... replace by: predicted.

Reviewer #4:

Remarks to the Author:

The authors isolate different haplotypes/ alleles of the Sr9 gene conferring resistance to the wheat

stem rust pathogen. The authors generated EMS mutants of multiple wheat lines carrying the known Sr9 alleles, selected for loss of Sr9 resistance. MuRenSeq then allowed the identification of the genes affected / associated with the loss of resistance phenotype ultimately leading to the isolation of the wt Sr9 alleles.

The manuscript is extremely complex and difficult to understand for probably most readers that lack significant background on cereal genetics and/or evolution of resistance genes (and those that have the background are actually authors on this manuscript). I suggest the authors to actually support their statements by showing the data in an appropriately labelled from in the figures to make the data accessible to the wide readership of this journal. Below some examples where the data seems missing/unclear.

-It will already greatly help to first of all provide an amino acid alignment with labelling the different domains etc of the WT Sr9 alleles. This will provide the reader with an overview of the similarities between the alleles and will make some other data redundant (Table S4). I know the mutant analysis led to the isolation of these sequences, but the whole description is difficult to follow.

-Can the authors mention in the text and the figures which cultivar is thought to carry which Sr9 allele. This would greatly help the reader to follow. For example, Gabo56 (Sr9h).

-I also miss any overview on the pathotypes of the Pgt races used. The authors mention the use of different races but it is unclear for the reader why which isolates were used. This information could be added to Table S1 and to me mentioned in the text (race XXX, avirulent on Sr9h,9e,9a).

In general, multiallelic resistance genes are thought to result from diversification to different pathogen races. I believe that some introduction to this effector recognition concept could be introduced here and discussed with respect to resistance gene diversification and allelic variation of the different Sr9 haplotypes alleles, in particular because the title of the manuscript implies data on resistance locus specificity.

I am not an expert on MuRenSeq, but from the EMS approach I would expect a number of mutants to not be affected in the Sr9 cds itself but in the promotor region and in genes required for Sr9 protein function. In particular chaperone such as SGT1 are generally required. How is it possible that the authors have not identified any of these mutations or say at least lines that do not have mutations in the Sr9 gene? Or have they but do not mention this?

Associated with this: Whilst I agree on the authors logic that this will facilitate the cloning of the functions Sr9 alleles, I am worried that the presentation of the data implies the AA changes found in the Sr9 allele of loss of function mutant lines are responsible for Sr9 loss of function. At this stage the authors cannot exclude that loss of function is not due to additional mutations in the genome, no of these lines. This should be clearly stated and the data presentation changed accordingly.

The Sr9 resistance phenotype is sometimes not all clear and apparently needs to be determined by pustule size instead of presence/absence of disease symptoms. In the current form I am not all convinced by the data presented for the Sr9h transgenic lines. I am unsure if the data itself is not solid, or if the presentation of the data is not. For example: only 50% of the Fielder T1 Sr9h lines are clearly more resistant to a Pgt race (that based on the data is avirulent on Sr9h, something the authors may actually want to clearly note in the manuscript text). Does this data really independently and by itself confirm Sr9h to be the resistance determinant?

With respect to the gene-for-gene model, I suggest the authors to include at least one virulent race and see if the enhanced immunity of the transgenic lines (and mutant cross lines) is truly due to Sr9 function to recognize a fungal AVR and not a pleiotropic effect of multi-copy NLR expression in transgenic lines. Because the transgenics are already available I consider these experiments feasible.

Statements

I disagree with the statement that multi-allelic resistance genes are rare in wheat. It doesn't appear rarer than in other species that are equally difficult to study. At this stage the community is biased on the genes used in common cultivars. A deeper look would probably lead to the identification of more

alleles of known resistance genes in all species. The authors may also wish to have a closer look at published work on multiple Pm2 alleles and include appropriate references to this in their manuscript. The authors consider the atypical CC domain of Sr21 and Sr9 a selling point of this manuscript. They even mention at some point (see below) that the CC domains of the resulting proteins and not (predicted) to be CC domains. This is not true, even the authors themselves show structural predictions of the proteins clearly highlighting the protein's CC domain. What I can extract in the authors Fig 4c, the 50 sth aa insertion when compared to other Sr proteins simply compromises a loop region located between the CC and the NB domain. It is therefore not all that surprising that expression of this domain in Nicotiana leads to cell death. Considering it is known that Sr9 is a functional resistance gene, I do not see if/how this inclusion of molecular data significantly contributes to the major points of this manuscript. For this data to be meaningful, one would need to analyses transgenics lacking the here so-called 'insertion' for function. The data may thus be moved to the supplement.

Clarification required / minor points:

- Fig S1a: mention in figure what Sr9 allele is present in which line.
 - Figure S2/ Line 150-154: These statements cannot be underlined by Figure S2 (contamination, early stop codons). I suggest a different/ clearer/better presentation of the different lines identified here. Do heterozygous SNPs suggest contamination? Please clarify these statements for the readers.
 - Figure 1a: Does Mutant M2 have G666E, P698D (SR9B) and S8266F (SR9E_H1), or does each mutant refer to each SR) allele? It may be best to actually label the background cultivar here which serves as reference for each SR9 WT. Also, please provide an alignment of the AA sequences derived from each SR9 allele.
 - Figure 1b: Details about the quality of the alphafold2 prediction missing (I assume it is alphafold2 not alphafold)
 - Figure 1 table. The table should not be part of a figure. Why including L6 but not other NLRs? Do other wheat NLRs / rust NLRs from other species generally have more LRRs than the Sr proteins? What point do the authors wish to make?
 - Lines 178 cont: is there a supp figure or table that supports these statements?
 - Figure S4: Can this data be presented like Fig 1A? The graphical information may help gathering the data more rapidly.
 - Line 211: why this isolate now?
 - Line 213: I assume the authors mean carrying Sr13 with a premature stop codon when mentioning 'lacking Sr13a'?
 - Figure 2: How do the authors conclude that TraesCS2B03G1225900 / Sr9 in CS (and the other cultivars indicated) is pseudogenes? Because of the early stop codon? If so, this fact should be mentioned earlier in the text or clarified in the Figure. The early stop codon is mentioned way later.
 - I presume the authors refer to alphafold2 throughout.
 - Line 220 cont.: This is not shown in Fig5, there is no Fig 5c. Is this data shown somewhere?
 - It is confusing that the authors refer to Table S1 whenever they discuss infection data. It may be sufficient to do so in a method section.
 - I cannot find any Nicotiana tabacum data anywhere but the authors mention this in the text
 - Line 344: how can the authors conclude that Sr9 does not have a CC domain? They show in Figure 4 that the CC aa sequence strongly aligns to other CC domains. It only has a AA insertion. When looking at the predicted structure this insertion is likely a loop region that adjuncts the predicted CC helices.
 - What is 12-in Table S6?
 - Line: 145 MutRenSeq 1451, something wrong with the reference
 - line 376 cont. 'We hypothesize that phenotypic differences between these two haplotypes may become evident when they are tested with a wider collection of Pgt races and as new races become available in the future.'
- This is somehow an un-scientific statement. I suggest to change to something like 'It is possibly that the two haplotypes differ in their function towards Pgt races that were not tested so far, or that one haplotype more rapidly evolves towards recognizing newly emerging race in the future.'

-line 379 cont: This change in Sr9b and Sr9h resulted in different specificities and presumably gave the latter allele the ability to recognise an avirulence gene in the Ug99 lineage (TTKSK). Similar to above, I think a change cannot 'give an ability' especially because the evolution of the difference is unclear. Change to sth similar to 'this difference between Sr9b and Sr9g is responsible for different recognition specificities and the ability to'

-Figure S9: on what basis were the NLR genes for this analysis chosen? The legend just says 'Sr and other NLR genes)

REVIEWER COMMENTS

Reviewer #1 (Remarks to the Author):

Zhang et al reported the isolation of five alleles at wheat stem rust resistance gene locus Sr9 located on chromosome 2B, namely Sr9b, Sr9g, Sr9e_h1, Sr9e_h2 and Sr9h, proved that the previously reported genes Sr9e and SrKN were actually two different haplotypes of Sr9e (Sr9e_h1 and Sr9e_h2). The authors also found these alleles encode a highly related nucleotide-binding site leucine-rich repeat (NB-LRR) type immune receptor, with an unreported N-terminal motif and an unusual long LRR domain. The protein encoded by the allele gene Sr21 from *T. monococcum* present on chromosome 2A also has the same characteristics like SR9 protein above, due to the close similarity between A and B subgenome. An interesting phenomenon that deserves further structural and functional exploration is that the two amino acid differences between SR9B, SR9G and SR9H, located in 26th LRR affects the resistance spectrum of wheat stem rust pathogen. This paper provides a better understanding of the evolution of wheat disease resistance allelic series.

The paper is in-depth in genomics research, but lacks further corresponding functional exploration. My recommendation would be to publish this paper after my specific comments are addressed.

Major issues:

1. Page 2 line 45: The sentence “that confers resistance to a unique spectrum of isolates of the wheat stem rust pathogen”. I think you should do some experiments to compare the spectrum against wheat stem rust pathogen between this NLR and common NLR.

To further enrich this manuscript, we performed additional experiments to help clarify the reviewer’s request:

Firstly, we validated the effectiveness and race-specificity of *Sr9h* in transgenic families *Sr9h-1a* and *Sr9h-5b* (refer to new added Figure S8). The two families showed significantly reduced average pustule sizes compared to Fielder for *Sr9h*-avirulent races TTKSK and RKRQC, but did not show significant differences from Fielder in response to *Sr9h*-virulent races QFCSC and MJGTC. Since the race-specificity of the *Sr9h* gene is confirmed in these experiments, we can conclude that *Sr9h* confers resistance to a unique spectrum of isolates of the wheat stem rust pathogen.

Secondly, although the phenotypic differences of the *Sr9* allelic series have been previously documented [The different resistance spectra of *Sr9a*, *Sr9b*, *Sr9d*, *Sr9e*, *Sr9f*, and *Sr9g* could be found in “Wheat Rusts: an atlas of resistance genes” by McIntosh et al. (1995). The information on *Sr9h* was mentioned in the paper (Rouse et al., 2014, Theor Appl Genet) that first characterized the *Sr9h* allele.], we performed an experiment to document these phenotypic differences and presented the data in the new added Table S8. In short, 12 different isolates of the wheat stem rust pathogen were inoculated onto monogenic lines possessing each of the *Sr9* alleles. Each allele possessed a unique race specificity. *Sr9f*, which we believe not to be a true allele of *Sr9*, but likely another closely linked gene, was described as effective to race 111x36 in another section of the manuscript but is ineffective to the isolates in Table S8 (refer to the new added result section: Resistance race specificities of SR9B, SR9G, and SR9H starting from line 319).

2. SR9H differs from SR9B and SR9G by only one and two residues, respectively, however only SR9H is resistant to Ug99. Here, it is necessary to substitute the residues M1169 and R1173 between SR9H, SR9B and SR9G, exchange the basic amino acids (arginine and lysine), and explore the resistance changes. In addition, it is not enough to show the polymorphic residues in LRR model, please analyze the structure and perform functional verification.

We understand the reviewer is interested in the functional verification to demonstrate the function of the two amino acids of SR9H. We feel this question could be perfectly answered by the two “natural” variants (SR9B and SR9G) of SR9H, which both conferred a different resistance spectrum. We can also perceive the potential suggestions by the reviewer for perhaps exploring how the two amino acids could affect the recognition process. We agree that if the corresponding pathogen effectors for SR9B, SR9G, and SR9H were known, it would be a perfect story to demonstrate how the resistance recognition/specificity has evolved. We would like to follow up on several of these research directions in future initiatives. Nonetheless, in order to clearly explore the resistance changes between SR9H, SR9B, and SR9G, we performed additional experiments and modified the manuscript accordingly:

The Sr9B, Sr9G, and Sr9H genes were present in the wheat lines ‘Chinese Spring’ (CS) chromosome substitution lines “CS/KF 2B”, “CS/Mq 2B”, and Fielder-background transgenic lines “Sr9h-1a” and “Sr9h-5b”. In order to demonstrate the resistance changes among these three alleles, they were each compared independently to the corresponding mutant lines (for Sr9B and Sr9G) and non-transgenic ‘Fielder’ background (for Sr9H). All materials, including additional controls CS and Sr9H genetic stock ‘RL6203’, were inoculated with six stem rust pathogen races (**new added Table S9**). Three of these races, QFCSC, RKRQC, and MGJTC, clearly discriminated the phenotypic effects of each allele. We included a summary figure showing a snapshot of the response of each line-race combination in the new added main Figure 4. For responses to these three races, we documented the average pustule sizes across leaves for each line-race combination and performed analyses of the mutants and transgenic lines compared to their respective controls (CS/KF 2B, CS/Mq 2B, and Fielder for Sr9B, Sr9G, and Sr9H, respectively). The analyses clearly demonstrated unique functional race-specificity for each allele (**new added Figure S8 and S12** where SR9B was only effective to race QFCSC, SR9G was only effective to race MGJTC, and SR9H was only effective to race RKRQC in addition to TTKSK). These results showed the differential functional effects of the residues M1169 and R1173 between SR9B, SR9G, and SR9H exhibited by race-specific resistance.

3. The prediction results of DeepCoil do not fully demonstrate the absence of CC domain in SR9 proteins, the model of SR21 and SR9H N-terminal domain looks similar to the structure of autoinhibited ZAR1 CC domain, would you provide a structure superposition and other stronger evidence. In addition, the SR50CC1-163-YFP and YFP proteins expressed in tobacco should also be detected by the immunoblot, and it would be better if the samples in Western blot could be on the one gel.

We added a supplementary figure showing the superposition of ZAR1/Sr35/YR5 and the SR9 N terminus (Figure S18). We also modified the figure showing the SR50CC1-163-YFP and the SR9CC & SR21CC auto-active fragments were all expressed and detected in one gel and added the result image of tobacco (**Figure S16c**).

Minor issues:

1. Page 3 line 71: “calcium channel” is inaccurate, ZAR1 resistosome is a calcium-

permeable cation channel.

Revised accordingly.

2. Page 3 line 72: “CryoEM” should be written as “Cryo-EM”.

Revised accordingly.

3. Page 3 line 73 and 75: delete the “-” in “NLR – binds” and “Anderson-with”.

Revised accordingly.

4. Page 5 line 138: “343-1,2,3,4,5,6, respectively (Table 1)”, I can’t find Table 1 in the manuscript, maybe it is Table S1.

Thanks for picking this up. It has been modified accordingly.

5. Page 7 line 219: I didn’t find Figure S5C in paper.

Thanks for picking this up. It was a wrong citation, it has now been revised accordingly.

6. Page 12 line 382: “N terminal” should be written as “N-terminal”.

We check through the whole manuscript and revised accordingly.

7. Please unify the following two formats, page 14 line 452: “10-15 days” and page 16 line 517: “2-5 d”

Revised accordingly.

8. Reference 8, 9, 12, 19, 20, 26 and 65: The citation formation about “Science (80-.)” is incorrect, please revise and check the rest of reference.

Revised accordingly.

9. Reference 16, 21, 22, 29, 60, 64 72, 74: Reference information is incomplete, please supplement and check all.

Revised accordingly.

10. Figure 2 and S7 legend last sentence: “The numbering I to IV highlights phylogenetic subgroups...” maybe it should be “The numbering I to V highlights phylogenetic subgroups...”.

Revised accordingly.

11. Figure S8a: The numbering I and II can be marked on the periphery of the phylogenetic grouping as in Figure 2 and S8.

Revised accordingly.

12. Table S3: The primer CT21CC_attb and CT9hCC_attb, change all lowercase letters to uppercase.

Revised accordingly.

13. Table S8: The footnote with the meaning of different colors in the table will be useful.

Revised accordingly. We removed the colour shading to avoid confusion.

Reviewer #2 (Remarks to the Author):

Review of Zhang et al NCOMMS-22-43739, Nov-2022

A single amino acid change can alter the specificity of the multi-allelic wheat stem rust resistance locus SR9

The authors open their study of the SR9 locus with a review of the fifteen-wheat stem rust resistance genes that have thus far been cloned. They point out that eleven of the fifteen encode a single class of protein, namely nucleotide-binding leucine-rich repeat (NLR) immune receptor proteins. The authors then launch into a brief description and characterization, including the discovery of each of the eight alleles, a thru h, that made up the original SR9 locus. As pointed out by the authors, Sr9c was dropped and later renamed Sr36 (BGRI). This, plus the more recent addition of Sr9h, left us with the seven

Sr9 alleles we know today.

As noted by the authors, the seven alleles of the SR9 locus have been deployed in agriculture to varying degrees. Alleles Sr9b, e, and g have been especially present in wheat cultivars. The most recently discovered Sr9h may also have a major impact because it is the only Sr9 allele that confers resistance to race TTKSK, or Ug99, and thus it has the potential to become an important wheat gene. The introduction has done a good job of describing the problem, briefly introducing the alleles, and making the reader aware of their importance.

KEY RESULTS: Procedures are similar to Zhang et al (2017). The alleles of the SR9 locus were sequenced and in doing so, there was more than one important fact established. First, six of the seven putative alleles were confirmed by molecular methods to be allelic, the exception being Sr9f whose status is left uncertain. The Sr9e allele was found to have two haplotypes, and possibly a third, which have thus far not been differentiated into separate alleles. Multiple haplotypes were not reported for any other allele. An interesting fact was that NLR clusters of these alleles are highly conserved, differing from each other by only one or two base-pairs. The conserved nature of the NLR clusters also allowed the authors to make perhaps the most unique conclusion of the study. They found when comparing the NLR clusters of the SR9 loci to that of the other fifteen cloned stem rust genes, that the Sr9 alleles and Sr21 are orthologous. While there have been many orthologous series of genes noted in wheat, to my knowledge, this is the first orthologous series established for stem rust resistance genes in wheat. The authors close the results section on the hope that publication of these gene sequences will be useful for studies of R-Avr (resistance-avirulence) interactions in plants.

VALIDITY: I don't find any flaws in the manuscript.

ORIGINALITY AND SIGNIFICANCE: The conclusions are definitively original and of immediate interest to people in my discipline and should be of interest to people in several other disciplines including plant pathologists, plant breeders, plant geneticists, general botanists, and general geneticists.

DATA AND METHODOLOGY: As noted earlier, the approach that the authors took with this study is similar to the approach of Zhang et al (2017), a study in which Rouse was also one of the lead scientists. I think the similarities in methodology and data presentation is fine. The reported methods are sufficiently detailed to enable reproduction of the results. Accession numbers are reported to enable testing of lines.

APPROPRIATE USE OF STATISTICS AND TREATMENT OF UNCERTAINTIES: I have no problem with the general use of statistics in the manuscript. However, I do have some concerns with the use of the Stakman scale (Stakman et al, 1962) for reading rust. Many readers will be unfamiliar with this scale. I am familiar with the scale so I can easily follow what the authors are trying to convey. But additional detail may be needed for readers unfamiliar with Stakman. Specifically, I think it would be useful for each plant to be marked as R or S. This has been done for some Figures, but not all. Take Figure 1a as an example. The authors mention in the legend that there is a mixture of R and S sibling plants in Figure 1A. Based on the numbering and infection types, it is easy enough to determine which are the sibling plants. But perhaps not so to an untrained eye, so identifying plants as R or S would be useful.

In discussing uncertainties and the Stakman scale, one Figure comes to mind, Figure S6. Based on the haplotypes shown in the legends, the two leaves to the left should be resistant, and they clearly are. The remaining 8 leaves should be susceptible, and seven are clearly susceptible, but the infection type shown for PI94674 is not in my opinion a clear susceptible

phenotype. Do the authors have any comments? Perhaps this is another case where identifying plants as R or S would be useful.

We revised the figure and figure legends accordingly with the labelling of R or S wherever suitable.

CONCLUSIONS: Yes, I find the conclusions robust, valid, and reliable.

SUGGESTED IMPROVMENTS: One item that I looked for in the manuscript was a comparison of the differences in the Sr9 alleles. The authors report that there is only a one or two base pair difference between these alleles, but these allelic differences are not listed in a table. The authors do report at the end of the manuscript that the allele sequences are available at the NCBI GenBank and they report the accession numbers. But it would be nice if a summary could be presented in a brief supplemental table. Perhaps there is a reason not apparent to me that this was not done. Basically, what I am arguing is expanding Table S4 to include all alleles.

We added a sequence alignment figure for all 5 proteins as a new added **Figure S6**.

EDITS: On line 145, “MutRenSeq1451” should be changed to “MutRenSeq14, 51”

Thanks for picking this up. It has been revised accordingly.

REFERENCES: All okay.

CLARITY AND CONTEXT: All okay

BGRI 2022. <https://globalrust.org/gene/sr36>

Stakman, E.C., Stewart, D.M. & Loegering, W.Q. (1962) Identification of physiologic races of *Puccinia graminis* var. *tritici*. USDA ARS E-617. US Government Printing Office, Washington, DC.

Zhang, W. et al. (2017) Identification and characterization of Sr13, a tetraploid wheat gene that confers resistance to the Ug99 stem rust race group. *Proc Natl Acad Sci U S A* 114, E9483–E9492.

Reviewer #3 (Remarks to the Author):

This paper describes the molecular identification of an allelic series of resistance genes against the wheat stem rust pathogen in wheat. The Sr9 alleles were identified and characterized using mutational analysis, genetic complementation by transformation and phylogenetic analysis based on genomic resources. The work reveals a series of genes that are very similar in sequence (some of them with only one or two amino acid polymorphisms), yet each of them providing a characteristic resistance spectrum to stem rust isolates. This makes the work highly valuable for understanding the molecular basis of specificity, although the recognized proteins (avirulence proteins) in the pathogen remain to be identified. There are currently no allelic series of resistance genes to rust known in the wheat genome, and there is only one well characterized allelic series against the mildew pathogen. This makes the study an original and novel contribution to understanding diversity of resistance genes in the globally important wheat crop. The work also identifies the Sr9 alleles and the Sr21 gene on the A genome as homoeologous genes, adding an important piece of data of relevance for the study of resistance gene evolution.

There are some shortcomings in the manuscript which mostly relate to data presentation, content of presented data and the depth of data analysis. Most importantly, there is a lack of clarity which makes the evaluation of the work difficult.

1. The abstract mentions that Sr9a was characterized in this study. However, there are no data

on Sr9a in the manuscript and it is unclear from the description of the work what is the reason of this.

The results for *Sr9a* and *Sr9d* characterization were presented in **lines 287-293**. They represent the most likely candidate genes for Sr9a and Sr9d. We reported that we confirmed only five SR9 alleles (*Sr9b*, *Sr9e_h1*, *Sr9e_h2*, *Sr9g*, and *Sr9h*) in the abstract instead of eight.

2. Figure 1a is unclear and needs a sister figure with sequence comparisons (aa sequence polymorphisms distinguishing the alleles; in addition to mutant sites given in the figure). The few aa polymorphisms that distinguish the different alleles (again: where is Sr9a?) must be clearly indicated so the reader gets an overview on the diversity, also related to the domain structure.

We would like to show in Fig. 1a the positions of different mutations and the amino acid changes caused by these mutations. We added a supplementary figure (**new added figure S6**) showing the protein sequence alignment of all the five identified haplotypes and labelled them with domain structures as the reviewer suggested. We also deposited both the genomic and the protein sequences into NCBI for readers who want to do more comparisons or analyses on them.

3. Where is Sr9d in Figure 1a?

Please refer to the answer to question 1.

4. At the bottom of Figure 1a there is a scheme that shows the diversity in the allelic sequences (down to 50% in the most C-terminal region?). This diversity is much higher than it is described in the text. Where do all the polymorphisms at the C-terminal end come from? Does this include paralogs, homoeologs, or Sr9a? This needs more clarity and a more detailed description in the text.

Thanks for picking this up. The major polymorphisms shown in Figure 1a resulted between SR9B, SR9G, SR9H group and the two SR9E group. We have included this statement in the text to make it clearer **“The major polymorphic region are between the SR9B, SR9G, SR9H cluster and the two SR9E alleles.”**

5. Figure 2 legend: Different SR9 alleles are shown. However, it is unclear what are functional alleles, what are paralogs. There is lack of clarity. It remains unclear how sequences that differ only in very few aa can give such differences in phylogenetic groupings. Or is this all relating to polymorphisms present in Sr9d and Sr9e (but their diversity is not addressed in the manuscript)?

We improved the figure legend for better clarity of details presented in the figure. The phylogenetic trees are based on DNA sequence. Most of the branch diversity in the phylogenetic grouping is represented by the alleles associated with *Sr9e* and *Sr9d*. The remaining alleles differ in minor changes on nucleotide as well as protein level and cluster as expected in proximity to each other. We labelled accessions carrying functional alleles in Figure 2a. In addition, we addressed the functional differences conferred by the SR9 alleles, with special attention to SR9G, SR9H, and SR9B through additional experiments described in the response to Reviewer #1 points #1 and #2.

6. The aa polymorphisms between the different allelic variants must be described in much more detail (see above comment 2). In the current version of the manuscript, it is not possible to fully understand the work based on the given information.

Please refer to the answer to question 2.

7. What is the unreported motif at their N termini mentioned in the abstract? This is not described any further in the manuscript.

This motif was described in lines 418 to 424 in the results section, figures S16 and S18, and lines 443-453 in the discussion section. We have also removed from the abstract the statement referring to the unreported motif at the N termini, given that the major thrust of the paper was defining the encoding sequences/variants of the multi-allelic *SR9* series.

8. There are very interesting observations on temperature sensitivity of different alleles. But this is not analyzed in depth, and it is unclear how this contributes to the study. The authors could study temperature dependence of autoactive versions of the allelic variants in transient expression assays in *Nicotiana* to verify that temperature differences are really caused by the NLRs themselves and do not relate to genetic background effects.

We have refrained from any in-depth study of temperature effects of these genes other than what has been reported in the literature. Our primary focus was the identification of the encoding genes. The *Sr9f* temperature effect study was to help validate the previously reported phenotype and in the case of *Sr9e_h2* to provide an example of the absence of temperature sensitivity of the phenotype within the temperature regimes of 18-25°C. We deleted two sentences from the short section of the text describing the temperature specificity of *SR9* alleles in order to emphasize only the unusual temperature specificity of *Sr9f* which differentiates it from the limited information known about *Sr21*, *Sr9a*, *Sr9b*, and *Sr9e*. Given the autoactive version is only the N terminal fragments of the SR9H/SR21, we consider this experiment would not be very informative regarding the temperature sensitivity.

9. It remains unclear how the experimental findings on the N-terminal domain contribute to the study. What are the conclusions of Figure S9 and Figure 4, and what is the purpose of these experiments?

N-terminal domain autoactivity has only been demonstrated in a few SR NLR proteins, e.g. SR50 and SR33, and both of the proteins actually fall into the MLA family. Given the unique features of the N terminus of SR21 and SR9H we discovered in this study, and the most similar proteins that have the unique structure is the YR5 BED-domain containing proteins that lost the N terminus autoactivity (as in the discussion), we feel it will be of potential interest for structural/molecular biologists, therefore worth reporting these results. It will be particularly useful information for evaluating the NLR type R protein structural conformation and signalling (for instance, models of interacting with the effector directly or indirectly) in future studies. We improved the text by adding the statement “This result could be potentially informative for studying the NLR type R protein structural conformation and signalling.” in the discussion section to clarify the result of this part of results.

Other comments:

Figure 1a: please indicate aa positions of the borders of the domains

We labelled the domain borders in the supplementary figure showing the alignment of all five proteins (new added Figure S6).

Figure 1 legend: ... nonsense mutations or premature stop codons... : These terms mean the same. It should be missense or premature stop codons.

Thanks for picking this up. Revised accordingly.

Supplementary figures:

Figure S2 contains information in unreadable small fonts. The figure must be edited and improved for clarity.

We separated Figure S2 into 4 independent figures (**Figures S2, S3, S4, and S5**) for a better resolution of each figure.

Figure S7b: The figure needs improvement for clarity. Sr21 was identified in DV92. Please indicate clearly which of the genes indicated in the figure of the DV92 genome is Sr21.

We labelled the Sr21 gene of DV92 in the plot and revised the figure legend to increase the clarity of the presented figure (**now as figure S14b**).

Related to Figure S7: Is anything known if the inversion of haplotype 2 has resulted in a known suppression of recombination at this locus in breeding programs?

We are not aware of suppressed recombination at this locus among lines that carry the inverted haplotype. However, the distal region of chromosome 2AS from some of these lines (Jagger, Mace, CDC Stanley) carry suppressed recombination due to the presence of the alien introgressed segment from *Aegilops ventricosa*.

Line 202: which Sr9 sequence?

We revised the sentence as “the *Sr9* sequence from Vernal emmer” to make it clearer.

There are few language problems in the manuscript and some editing is necessary: e.g. L65: ... and most of them have a predictable... replace by: predicted.

Revised accordingly.

Reviewer #4 (Remarks to the Author):

The authors isolate different haplotypes/ alleles of the Sr9 gene conferring resistance to the wheat stem rust pathogen. The authors generated EMS mutants of multiple wheat lines carrying the known Sr9 alleles, selected for loss of Sr9 resistance. MuRenSeq then allowed the identification of the genes affected / associated with the loss of resistance phenotype ultimately leading to the isolation of the wt Sr9 alleles.

The manuscript is extremely complex and difficult to understand for probably most readers that lack significant background on cereal genetics and/or evolution of resistance genes (and those that have the background are actually authors on this manuscript). I suggest the authors to actually support their statements by showing the data in an appropriately labelled from in the figures to make the data accessible to the wide readership of this journal. Below some examples where the data seems missing/unclear.

-It will already greatly help to first of all provide an amino acid alignment with labelling the different domains etc of the WT Sr9 alleles. This will provide the reader with an overview of the similarities between the alleles and will make some other data redundant (Table S4). I know the mutant analysis led to the isolation of these sequences, but the whole description is difficult to follow.

We provided a new supplementary figure (**new added Figure S6**) showing the alignment of all five proteins with the labelling of the domain structure.

-Can the authors mention in the text and the figures which cultivar is thought to carry which Sr9 allele. This would greatly help the reader to follow. For example, Gabo56 (Sr9h).

Revised accordingly in both the text and the figure as suggested by the reviewer.

-I also miss any overview on the pathotypes of the Pgt races used. The authors mention the use of different races but it is unclear for the reader why which isolates were used. This information could be added to Table S1 and to me mentioned in the text (race XXX, avirulent on Sr9h,9e,9a).

Table S1 has been revised and added the information (virulent and avirulent pattern of each race) the reviewer suggested.

In general, multiallelic resistance genes are thought to result from diversification to different pathogen races. I believe that some introduction to this effector recognition concept could be introduced here and discussed with respect to resistance gene diversification and allelic variation of the different Sr9 haplotypes alleles, in particular because the title of the manuscript implies data on resistance locus specificity.

Thanks for the reviewer's suggestion to improve this manuscript, we revised the text according to the reviewer's suggestion (line 125 to 128, page 4).

I am not an expert on MuRenSeq, but from the EMS approach I would expect a number of mutants to not be affected in the Sr9 cds itself but in the promotor region and in genes required for Sr9 protein function. In particular chaperone such as SGT1 are generally required. How is it possible that the authors have not identified any of these mutations or say at least lines that do not have mutations in the Sr9 gene? Or have they but do not mention this? Associated with this: Whilst I agree on the authors logic that this will facilitate the cloning of the functions Sr9 alleles, I am worried that the presentation of the data implies the AA changes found in the Sr9 allele of loss of function mutant lines are responsible for Sr9 loss of function. At this stage the authors cannot exclude that loss of function is not due to additional mutations in the genome, no of these lines. This should be clearly stated and the data presentation changed accordingly.

The reviewer is correct that the non-synonymous change may occur in regions other than the CDS of the target gene. We also had previously experienced that we could not find any SNPs in 1 or 2 of the mutants in some of our other gene targets. This was the case with *Sr9g* as we only identified 3 mutants out of 5 that carried SNPs (Figure S4). We equally share the curiosity of the reviewer regarding the inadequacy of finding second site mutations that function in Sr-mediated resistance.

Indeed, the majority of mutants that we have identified for numerous *R* genes as well as those from other wheat researchers have possessed non-synonymous mutations in corresponding causal genes that were validated by transgenics : *Sr9h* (this study), *Sr26* and *Sr61* (Zhang et al., 2021), *Yr5a*, *Yr5b*, and *Yr7* (Marchal et al., 2018), *Pm1* (Hewitt et al., 2021), *Lr67* (Moore et al., 2015), *Sr13* (Zhang et al., 2017) etc. The possible explanations we postulated include:

a. The generating of, and screening standards of mutants —we optimised the mutant screen experiment by selecting the most suitable rust isolate that the knockout mutant would clearly give a completely susceptible response and we excluded any mutants that showed a “knockdown” or “reduced resistance” type of response, as well as mutants with some other growth defect traits-such as stunting etc., which we observed in our mutant screening all the time. We assume this type of screening would exclude some of the mutants that may be caused by chaperones or other genes that may require resistance expression regulation.

b. Possible genome redundancy of genes required for resistance/defence signalling in contrast to unique primary recognition receptors specific to the rust race.

We have improved the text from line 157 to 162 in page 5 to reflect these cases.

The Sr9 resistance phenotype is sometimes not all clear and apparently needs to be

determined by pustule size instead of presence/absence of disease symptoms. In the current form I am not all convinced by the data presented for the Sr9h transgenic lines. I am unsure if the data itself is not solid, or if the presentation of the data is not. For example: only 50% of the Fielder T1 Sr9h lines are clearly more resistant to a Pgt race (that based on the data is avirulent on Sr9h, something the authors may actually want to clearly note in the manuscript text). Does this data really independently and by itself confirm Sr9h to be the resistance determinant?

With respect to the gene-for-gene model, I suggest the authors to include at least one virulent race and see if the enhanced immunity of the transgenic lines (and mutant cross lines) is truly due to Sr9 function to recognize a fungal AVR and not a pleiotropic effect of multi-copy NLR expression in transgenic lines. Because the transgenics are already available I consider these experiments feasible.

Thank you for the comments. We performed additional experiments and additional analyses of previous experiments and modified the manuscript accordingly (line 259 to 273):

1. We analyzed the pustule size data across the transgenic lines compared to the background 'Fielder' and presented this in **Figure S7**. From each of the five transgenic *Sr9h* families, the average pustule sizes of infected leaves from plants that were PCR-confirmed to possess the transgene were compared to the 'Fielder' background. All five families showed significantly reduced pustule sizes compared to 'Fielder' ($p < 0.001$). A total of six plants across three transgenic families did not possess the *Sr9h* transgene (**Figure S7**). We also compared the average pustule size of these progeny without *Sr9h* compared to Fielder and found no significant difference.
2. We validated the effectiveness and race-specificity of *Sr9h* in transgenic families *Sr9h-1a* and *Sr9h-5b* (**Figure S8**). The two transgenic families showed significantly reduced average pustule sizes compared to Fielder for *Sr9h*-avirulent races TTKSK and RKRQC, but did not show significant differences from Fielder in response to *Sr9h*-virulent races QFCSC and MJGTC. Since the race-specificity of the *Sr9h* gene is confirmed in these experiments, we can conclude that the enhanced resistance of the transgenic lines is truly due to *Sr9h*-specific function, and not just a pleiotropic effect of multi-copy NLR expression in the transgenic lines.

Statements

I disagree with the statement that multi-allelic resistance genes are rare in wheat. It doesn't appear rarer than in other species that are equally difficult to study. At this stage the community is biased on the genes used in common cultivars. A deeper look would probably lead to the identification of more alleles of known resistance genes in all species. The authors may also wish to have a closer look at published work on multiple Pm2 alleles and include appropriate references to this in their manuscript.

The reviewer may be right and at best we can only go with what is currently known in wheat when we take into account over 300 catalogued R gene loci for biotrophic pathogens. Here, the rarity refers to as a proportion of total effective genes (ie ~ 60 Sr loci but only 5 of them have documented alleles, *SR7*, *SR8*, *SR9*, *SR13*, and *SR22*). This rarity applies again as 1/80 for Yr, 3/80 for Lr, and 3/60 for Pm approximately. We modified this statement in the abstract to clarify this point.

The authors consider the atypical CC domain of Sr21 and Sr9 a selling point of this manuscript.

They even mention at some point (see below) that the CC domains of the resulting proteins and not (predicted) to be CC domains. This is not true, even the authors themselves show structural predictions of the proteins clearly highlighting the protein's CC domain. What I can extract in the authors Fig 4c, the 50 sth aa insertion when compared to other Sr proteins simply compromises a loop region located between the CC and the NB domain. It is therefore not all that surprising that expression of this domain in Nicotiana leads to cell death. Considering it is known that Sr9 is a functional resistance gene, I do not see if/how this inclusion of molecular data significantly contributes to the major points of this manuscript. For this data to be meaningful, one would need to analyses transgenics lacking the here so-called 'ionsertion' for function. The data may thus be moved to the supplement.

The modelling of the insertion regions was highlighted in cyan and showed as a-helix structure instead of a loop region (original Figure 4c, now as new figure S16c). The 51 aa insertions or atypical N terminal regions were highlighted in this manuscript mainly for the potential interest of structural biologists.

We added some more experiments and data in this revision and moved this part of the results into supplementary as suggested by the reviewer.

Clarification required / minor points:

-Fig S1a: mention in figure what Sr9 allele is present in which line.

Revised figure accordingly.

-Figure S2/ Line 150-154: These statements cannot be underlined by Figure S2 (contamination, early stop codons). I suggest a different/ clearer/better presentation of the different lines identified here.

Do heterozygous SNPs suggest contamination? Please clarify these statements for the readers.

We improved the text wording to make it clearer (Line 157-162).

-Figure 1a: Does Mutant M2 have G666E, P698D (SR9B) and S8266F (SR9E_H1), or does each mutant refer to each SR) allele? It may be best to actually label the background cultivar here which serves as reference for each SR9 WT. Also, please provide an alignment of the AA sequences derived from each SR9 allele.

Thanks for picking up this error. M2 P698S should be M1 P698S. We revised the figure and legend to make it clearer as suggested. We added a new **Figure S6** to show the sequence alignment of all 5 SR9 proteins.

-Figure 1b: Details about the quality of the alphafold2 prediction missing (I assume it is alphafold2 not alphafold)

Revised accordingly. Prediction confidence information was added in new **Figure S19**.

-Figure 1 table. The table should not be part of a figure. Why including L6 but not other NLRs? Do other wheat NLRs / rust NLRs from other species generally have more LRRs than the Sr proteins? What point do the authors wish to make?

We moved the table to the supplementary **Table S3**. L6 was included as a comparison between CNL and TNLs. The result we show here is to highlight the large variation of LRR numbers within the cloned SRs, as the number of LRR would apparently affect the whole protein folding and 3D structure confirmation and these confirmation changes may lead to improving our knowledge of R-Avr recognition interface/site, which we believe in the LRR region.

-Lines 178 cont: is there a supp figure or table that supports these statements?

We added the result data in the corresponding paragraphs (**lines 194-200 and lines 204-209**).

-Figure S4: Can this data be presented like Fig 1A? The graphical information may help gathering the data more rapidly.

I think the reviewer means Table S4, part of this figure was illustrated in the new supplementary **Figure S6**.

-Line 211: why this isolate now?

I assume the question is why we used TRTTF. TRTTF was the race that we used to find the Sr9e_KN gene in the first place (as previously described in Li et al. 2021). We found the SR13 mutant lines were resistant to TRTTF. So we mapped the gene with TRTTF, and then validated the transgenics with TRTTF. We modified the text for a better flow.

-Line 213: I assume the authors mean carrying Sr13 with a premature stop codon when mentioning 'lacking Sr13a'?

We revised the text to make it clearer.

-Figure 2: How do the authors conclude that TraesCS2B03G1225900 / Sr9 in CS (and the other cultivars indicated) is pseudogenes? Because of the early stop codon? If so, this fact should be mentioned earlier in the text or clarified in the Figure. The early stop codon is mentioned way later.

Revised accordingly. We moved the early stop codon statement to the beginning of the paragraph (**line 295-298**).

-I presume the authors refer to alpafold2 throughout.

Revised accordingly.

-Line 220 cont.: This is not shown in Fig5, there is no Fig 5c. Is this data shown somewhere?

Thanks for picking up this error, it should be **Figure S1c**.

-It is confusing that the authors refer to Table S1 whenever they discuss infection data. It may be sufficient to do so in a method section.

Revised accordingly. We removed some of the citation of Table S1.

-I cannot find any Nicotiana tabacum data anywhere but the authors mention this in the text

The tobacco phenotype figure was added in the new Figure S16.

-Line 344: how can the authors conclude that Sr9 does not have a CC domain? They show in Figure 4 that the CC aa sequence strongly aligns to other CC domains. It only has a AA insertion. When looking at the predicted structure this insertion is likely a loop region that adjuncts the predictedCC helices.

As we responded previously, the modelling of the insertion regions was highlighted in cyan and showed as a-helix structure instead of a loop region (**new Figure S16c, S18**).

We agree we could not conclude that the Sr9 does not have a Coiled-Coil domain until the crystal structure is solved as all the modelling are only predictions. Therefore we modified the text wording and moved all this part of result into supplementary data.

-What is 12-in Table S6?

"12-" is the phenotyping score for stem rust. We modified the table (now as **Table S7**) footnote to reflect if multiple seedling infection types were observed on the same leaf, then all infection types were listed in order of frequency of occurrence.

-Line: 145 MutRenSeq 1451, something wrong with the reference

Revised accordingly.

-line 376 cont. 'We hypothesize that phenotypic differences between these two haplotypes may become evident when they are tested with a wider collection of Pgt races and as new races become available in the future.'

This is somehow an un-scientific statement. I suggest to change to something like 'It is possibly that the two haplotypes differ in their function towards Pgt races that were not tested so far, or that one haplotype more rapidly evolves towards recognizing newly emerging race in the future.'

Revised accordingly (Line 436 to 438) .

-line 379 cont: This change in Sr9b and Sr9h resulted in different specificities and presumably gave the latter allele the ability to recognise an avirulence gene in the Ug99 lineage (TTKSK).

Similar to above, I think a change cannot 'give an ability' especially because the evolution of the difference is unclear. Change to sth similar to 'this difference between Sr9b and Sr9g is responsible for different recognition specificities and the ability to

Revised accordingly (Line 440 to 441) .

-Figure S9: on what basis were the NLR genes for this analysis chosen? The legend just says 'Sr and other NLR genes)

We revised the figure legend and added the information. These proteins are all NLR type R proteins from other plant species, it was sourced from one of our previous studies in Zhang et al. 2021, *Nat Commun*.

Reviewers' Comments:

Reviewer #1:

Remarks to the Author:

The authors have fully addressed my questions and made corresponding modification. For this version, I have no more questions and the paper is now suitable for publication.

Reviewer #2:

Remarks to the Author:

As the manuscript has already gone through a round of review, I will not add major comments or changes other than to state that I still find the manuscript well worth publishing in Nature Communications. Therefore, I will limit my comments concerning the merits of the manuscript. The authors have addressed the change that I suggested. These changes and the changes suggested by the other reviewers have strengthened the manuscript. However, I do have additional minor changes to suggest, mostly of formatting, spelling, and other minor suggestions. They are as follows:

Line 60 – Sixteen genes are listed with the addition of Sr43, so change Fifteen to Sixteen.

Line 298 and others. Chinese Spring is written in full on this line, but the initials CS is defined for Chinese Spring on line 141. Chinese Spring is first mentioned on line 100, but CS is not defined here. There are other mentions of Chinese Spring rather than the initials after line 298. The authors need to verify the proper use of the initials CS throughout the document.

Figure S3 and Table S2. Vernstein is spelled as Verstein
Figure S18.

In the text, ZAR1 is discussed. In Fig S18, it looks like ZAR1 is identified as ZARI. It is correct in the legend for Fig S18, only in the actual figure is it questionable. Perhaps this is only a font issue. Please check to see if it needs correction in the figure.

Table S9 footnote: indicating is misspelled.

References List

References 41 and 53 – correct capitalization of words in title

Check italicization in References 18 (*Aegilops Sharonensis*), 36 (*Triticum timopheevi*), 40 (Sr9), 65 (Sr33) and 74 (*Triticum aestivum*)

Reference 58, the volume number is missing.

In references, volume numbers are usually in bold, but occasionally are not (eg reference 53).

Also, remove issue number in reference 53.

Reference 71 – volume number and page numbers are missing.

Reference 81- capitalize plant

Based on the number of suggestions I had for the references, there are probably others that a more thorough check would find.

Reviewer #3:

Remarks to the Author:

The authors have revised the manuscript and present a much improved version. They have addressed all my concerns.

I have only two minor suggestions in relation to my comments on the original manuscript:

point 1 and 3: It is mentioned that Sr9a and Sr9d have between 98.8 and 99.7 DNA sequence identity to the Chinese Spring sequence. It would be interesting to present these sequences in the paper or at least mention, where the polymorphisms occur.

point 4: Refer to Figure S6 after the new sentence.

Reviewer #4:

Remarks to the Author:

The authors have sufficiently incorporated additional data from new experiments, primarily within the supplemental figures, and have revised the text with the intent of addressing the points I had previously raised in my review.

Point by point response to the REVIEWERS' COMMENTS

Reviewer #2 (Remarks to the Author):

As the manuscript has already gone through a round of review, I will not add major comments or changes other than to state that I still find the manuscript well worth publishing in Nature Communications. Therefore, I will limit my comments concerning the merits of the manuscript. The authors have addressed the change that I suggested. These changes and the changes suggested by the other reviewers have strengthened the manuscript. However, I do have additional minor changes to suggest, mostly of formatting, spelling, and other minor suggestions. They are as follows:

Line 60 – Sixteen genes are listed with the addition of Sr43, so change Fifteen to Sixteen.

Revised accordingly.

Line 298 and others. Chinese Spring is written in full on this line, but the initials CS is defined for Chinese Spring on line 141. Chinese Spring is first mentioned on line 100, but CS is not defined here. There are other mentions of Chinese Spring rather than the initials after line 298. The authors need to verify the proper use of the initials CS throughout the document.

Revised accordingly.

Figure S3 and Table S2. Vernstein is spelled as Verstein

Revised accordingly.

Figure S18.

In the text, ZAR1 is discussed. In Fig S18, it looks like ZAR1 is identified as ZARI. It is correct in the legend for Fig S18, only in the actual figure is it questionable. Perhaps this is only a font issue. Please check to see if it needs correction in the figure.

Revised accordingly.

Table S9 footnote: indicating is misspelled.

Revised accordingly.

References List

References 41 and 53 – correct capitalization of words in title

Check italicization in References 18 (*Aegilops Sharonensis*), 36 (*Triticum timopheevi*), 40 (Sr9), 65 (Sr33) and 74 (*Triticum aestivum*)

Reference 58, the volume number is missing.

In references, volume numbers are usually in bold, but occasionally are not (eg reference 53.

Also, remove issue number in reference 53.

Reference 71 – volume number and page numbers are missing.

Reference 81- capitalize plant

Based on the number of suggestions I had for the references, there are probably others that a more thorough check would find.

We checked and revised the reference list again accordingly.

Reviewer #3 (Remarks to the Author):

The authors have revised the manuscript and present a much improved version. They have addressed all my concerns.

I have only two minor suggestions in relation to my comments on the original manuscript:

point 1 and 3: It is mentioned that Sr9a and Sr9d have between 98.8 and 99.7 DNA sequence identity to the Chinese Spring sequence. It would be interesting to present these sequences in the paper or at least mention, where the polymorphisms occur.

We have now added at line 292-295 the sentence "Sequence comparisons of the candidate SR9A protein with the validated SR9 alleles, for example SR9B, showed amino acid substitution differences occurred predominantly in the LRR region, while SR9D differences were spread throughout the protein (Figure S6)."

point 4: Refer to Figure S6 after the new sentence.

Revised accordingly.